

# Modelling supraglacial debris-cover evolution from the single glacier to the regional scale: an application to High Mountain Asia

Loris Compagno[1,2], Matthias Huss[1,2,3], Evan Stewart Miles[2], Michael James McCarthy[2], Harry Zekollari[4,5,1,2], Francesca Pellicciotti[2], and Daniel Farinotti[1,2]

[1]Laboratory of Hydraulics, Hydrology and Glaciology (VAW), ETH Zurich, Zurich, Switzerland.
[2]Swiss Federal Institute for Forest, Snow and Landscape Research (WSL), Birmensdorf, Switzerland.
[3]Department of Geosciences, University of Fribourg, Fribourg, Switzerland.
[4]Department of Geoscience and Remote Sensing, Delft University of Technology, Netherlands.
[5]Laboratoire de Glaciologie, Université libre de Bruxelles, Belgium.

**Correspondence:** Loris Compagno (compagno@vaw.baug.ethz.ch)

**Abstract.** Currently, about 12-13% of High Mountain Asia's glacier area is debris-covered, altering its surface mass balance. However, in regional-scale modelling approaches, debris-covered glaciers are typically treated as clean-ice glaciers, leading to a potential bias when modelling their future evolution. Here, we present a new approach for modelling debris area and thickness evolution, applicable from single glaciers to the global scale. We implement the module into the Global Glacier

Evolution Model (GloGEMflow), a combined mass-balance ice-flow model. The module is initialized with both glacier-specific observations of the debris' spatial distribution and estimates of debris thickness, accounts for the fact that debris can either enhance or reduce surface melt depending on thickness, and enables representing the spatio-temporal evolution of debris extent and thickness. We calibrate and evaluate the module on a selected subset of glaciers, and apply the model using different climate scenarios to project the future evolution of all glaciers in High Mountain Asia until 2100. Compared to 2020, total

glacier volume is expected to decrease by between $35\pm15\,\%$ and $80\pm11\,\%$, which is in line with projections in the literature. Depending on the scenario, the mean debris-cover fraction is expected to increase, while mean debris thickness is modelled to show only minor changes, albeit large local thickening is expected. To isolate the influence of explicitly accounting for supraglacial debris-cover, we re-compute glacier evolution without the debris-cover module. We show that glacier geometry, area, volume and flow velocity evolve differently, especially at the level of individual glaciers. This highlights the importance

of accounting for debris-cover and its spatio-temporal evolution when projecting future glacier changes.

## 1   Introduction

In High Mountain Asia (HMA), debris-covered and clean-ice glaciers are losing mass due to climate change (Brun et al., 2017; Zemp et al., 2019; Shean et al., 2020; Hugonnet et al., 2021). Since the atmosphere is expected to warm further (IPCC, 2019), more glacier mass is expected to be lost (Marzeion et al., 2020; Rounce et al., 2020). Understanding how sensitive HMA

glaciers are to changes in climate is crucial to quantify the future glacier evolution in the area.



A key unknown is the present and future influence of supraglacial debris cover in moderating melt rates for the 12-13% of HMA's glacier area that is presently covered by debris (Herreid and Pellicciotti, 2020). A better understanding is necessary to accurately predict future water availability, to assess impacts on irrigation, hydropower, and both public and private usage of water (Biemans et al., 2019; Farinotti et al., 2019b; Fyffe et al., 2019; Immerzeel et al., 2020; Miles et al., 2021), to anticipate hotspots of hazards such as ice-dammed or proglacial lakes (Emmer et al., 2014; Zheng et al., 2021), or to project the glaciers' contribution to sea-level rise (Edwards et al., 2021).

Debris cover has a considerable effect on ice melting (Nicholson et al., 2018; Rounce et al., 2021). If debris is thinner than a few centimetres, it enhances ice melt due to darkening of the glacier surface (Owen et al., 2003; Anderson and Anderson, 2016). If debris is thicker than a few centimetres, instead, it reduces ice melt due to its insulating proprieties (Østrem, 1959; Reznichenko et al., 2010; Rounce et al., 2021).

Since glaciers are presently in disequilibrium with climate (Marzeion et al., 2018; Zekollari et al., 2020; Miles et al., 2021), their debris-covered areas are evolving through time (Stokes et al., 2007; Bhambri et al., 2011; Bolch et al., 2011; Shukla and Qadir, 2016; Tielidze et al., 2020), as changes in debris extent are linked to the glaciers' mass balance (Mölg et al., 2019). Indeed, medial moraines and debris patches – which are formed by the accumulation and transport of debris – tend to grow and to expand laterally with increasing ablation (Anderson, 2000; Jouvet et al., 2011; Rowan et al., 2015; Kienholz et al., 2017; Wirbel et al., 2018; Verhaegen et al., 2020). Additionally, ice-marginal moraines that may become unstable when glaciers retreat, can supply the ice surface with additional debris (Van Woerkom et al., 2019). As a consequence, glaciers with negative mass balances tend to increase their debris-cover fractions through time (Stokes et al., 2007; Bhambri et al., 2011; Bolch et al., 2011; Shukla and Qadir, 2016; Tielidze et al., 2020). An exception is given by the Karakoram region, where positive and negative debris-cover changes offset one another during the past 40 years (Herreid et al., 2015). This is most probably the consequence of the neutral, or even slightly positive mass balance in the region (Gardelle et al., 2013; Farinotti et al., 2020).

For glaciers with negative mass balances, the debris also progressively expands up-glacier (Stokes et al., 2007), together with the rise of the equilibrium line altitude (ELA). Indeed, although the presence of debris reduces ice melt at the glacier's lowest elevations, melting can be stronger on the debris-free ablation surface higher up. This fosters the expansion of the debris-cover fraction through the melt-out of englacial debris transported by the glacier's ice flow (Stokes et al., 2007; Rowan et al., 2015).

By combining estimates of sub-debris melt with surface temperature inversion methods, Rounce et al. (2021) recently presented the first global estimate of supraglacial debris thickness on glaciers. The estimate refers to about 2008, but debris thickness evolves through time. The few direct observations available indicate a debris-cover thickening in the last decades (e.g. Gibson et al., 2017; Verhaegen et al., 2020), most probably related to the negative mass balances induced by ongoing climate change as well as to the resulting glacier thinning and decelerated ice flow (Verhaegen et al., 2020; Anderson et al., 2021). Supraglacial ice cliffs and ponds might additionally contribute to this, as they enhance local ablation of debris-covered glaciers (Sakai et al., 2000, 1998; Ragettli et al., 2016; Miles et al., 2018) and evolve as well (Narama et al., 2017; Watson et al., 2017; Chand and Watanabe, 2019; Buri et al., 2021; Ferguson and Vieli, 2021).

Regional and global models with various levels of complexity have been used to simulate the HMA's future glacier evolution (see Marzeion et al., 2020, for a model inter-comparison). The models use different methodologies for computing ablation, ac-



cumulation, or geometry changes, but rarely take into account the debris cover and its spatio-temporal evolution. An exception is the study by Kraaijenbrink et al. (2017) that presented the first HMA glacier projections by explicitly accounting for the effect of supraglacial debris. However, neither did the study consider an evolution of debris extent and thickness in the future, nor did it model ice flow explicitly or use glacier-specific mass balance data for calibration. Glacier-specific studies considering debris-cover evolution exist (e.g. Jouvet et al., 2011; Rowan et al., 2015; Kienholz et al., 2017; Scherler and Egholm, 2020; Verhaegen et al., 2020), as well as theoretical and process based modelling studies (Anderson and Anderson, 2016; Ferguson and Vieli, 2021) but the majority are based on higher-order ice-flow models and require rather extensive observational data. Thus, the corresponding methods are hardly applicable at larger scales.

Here, we present a new debris area and thickness evolution module applicable to both individual glaciers and the regional to global scale. The module is included into the Global Glacier Evolution Model (GloGEMflow), a combined mass-balance (Huss and Hock, 2015) ice-flow (Zekollari et al., 2019) model. We extensively calibrate and evaluate the debris-cover module, and showcase its applicability for all glaciers in High Mountain Asia. We focus on the future evolution of debris cover, and determine the impacts that explicitly modelling debris cover evolution has on transient glacier evolution. To do so, we model all HMA glaciers between 2000 and 2100. The modelling is based on five shared socioeconomic pathways (SSP119, SSP126, SSP245, SSP370 and SSP585) from the sixth phase of the Coupled Model Intercomparison Project (CMIP6) and the results are compared to model runs that do not explicitly account for debris cover (Marzeion et al., 2020; Edwards et al., 2021). We discuss the resulting differences in terms of glacier mass balance, glacier evolution, and ice-flow velocity, which allows us to assess the importance of accounting for debris-cover evolution in regional studies.

## 2   Data

To model the evolution all 95'536 glaciers in HMA over the 21st century, different data sets are used (see Fig. 1).

### 2.1   Glacier Geometry

We use glacier outlines of the Randolph Glacier Inventory version 6.0 (RGI 6.0, RGI Consortium, 2017), which is a global inventory of glacier outlines. For HMA glaciers, the RGI outlines are based on remote sensing data acquired between 1998 and 2013. For the ice thickness, we use the consensus estimate by Farinotti et al. (2019a), which is based on an ensemble of models using characteristics of the glacier surface (e.g. slope and surface velocities) and principles of ice flow dynamics for ice thickness inversion. For the modelling, the geometry is condensed by subdividing each glacier into elevation bands of 10 m, including tributary glaciers (Huss and Hock, 2015), i.e. they are not treated separately. In this elevation-dependent representation, the transversal glacier bed shape is parametrized assuming a glacier cross-section that has the form of an isosceles trapezoid with 45° base angles (see Zekollari et al., 2019, for more details).



**Figure 1.** (**a**) Extent of HMA glaciers (white) as per Randolph Glacier Inventory version 6 (RGI Consortium, 2017). The three main RGI regions (Central-Asia, South-Asia-West, and South-Asia-East) are shown by blueish, reddish, and greenish colours, respectively. RGI second-order regions are labelled individually. Three glaciers are highlighted to illustrate glacier-specific model results (red spheres with numbers). (**b** & **c** & **d**) Map of the three highlighted glaciers with their mean 2000-2016 debris thickness given by colours (scale in panel **b**). Glacier outlines and debris thickness are from RGI Consortium (2017) and McCarthy et al. (sub.), respectively. For each glacier, $V$ is the glacier ice volume according to Farinotti et al. (2019a), $A$ is the glacier area according to RGI 6.0, $A_{\mathrm{debris}}$ is the debris-covered area, and $h_{\mathrm{debris}}$ is the mean debris-cover thickness according to McCarthy et al. (sub.). (**e** & **f** & **g**) Glacier hypsometry (area per 10 meter elevation band) and debris-covered area distribution at inventory date. $n$ is the number of glaciers within each region (RGI Consortium, 2017) Map source: Natural Earth.



## 2.2 Debris cover and Østrem curves

For each glacier with an area >2 km$^2$ and that has not been identified as a surging glaciers in RGI 6.0 (i.e. 4863 glaciers in total, see RGI Consortium, 2017), the debris coverage is represented by a debris-cover mask generated using Landsat scenes acquired between 2013 and 2017 (Scherler et al., 2018), as well as spatially distributed debris-cover thickness maps and glacier-specific Østrem curves (i.e. a function that characterizes the relation between debris-cover thicknesses and melt rates Østrem, 1959).

The debris thickness maps are based on McCarthy et al. (sub.), and were obtained through a simplified surface mass-balance inversion procedure similar to (Ragettli et al., 2015; Rounce et al., 2018). In this procedure, Østrem curves were created for each glacier using an energy-balance model for debris-covered glaciers, then inverted using 'observation'-based surface mass balance data from Miles et al. (2021). The energy-balance model was initiated at randomly chosen points on the surface of each glacier. The surface mass balance at these points was then computed for a large set of arbitrary debris thicknesses and debris properties. Østrem curves were then fitted to these surface mass balance values by using:

$$b = \frac{i_{\mathrm{debris}} \cdot k_{\mathrm{debris}}}{h + k_{\mathrm{debris}}} \qquad (1)$$

where $b$ is the local surface mass balance, $i_{\mathrm{debris}}$ and $k_{\mathrm{debris}}$ are free parameters, and $h$ is the debris thickness (m). Comparison to surface mass balance data inferred by Miles et al. (2021) finally allowed determining the actual debris thickness. The so obtained information represent the supraglacial debris conditions for the 2000-2016 period (McCarthy et al., sub.). The spatially-distributed debris-cover information is divided into elevation bands of 10 m, in line with the approach used for the ice thickness. The Østrem curves are instead directly integrated into our mass balance module (see section 3.1). We used this new dataset, rather than the dataset of Rounce et al. (2021), because it was generated using state-of-the-art input data sets and modelling techniques, and was more thoroughly validated for HMA.

To calibrate and evaluate the debris-evolution module (see 3.2 section), we use Hexagon and multiple Landsat satellite images acquired between 1973-1976 and 1987-2019, respectively. Since the Hexagon images are monochromatic, debris cover has been delineated manually for 31 glaciers distributed through HMA. We use glaciers with sufficient contrast between debris and clean ice, little to absent shadows on the glacier surface, and without snow on the ablation zone. For the Landsat satellite images, debris is identified automatically following the multi-date composite approach of Scherler et al. (2018). We apply this method to the combined multi-sensor Landsat archive in Google Earth Engine for additional epochs with sufficient Landsat acquisitions: 1987-1991, 1994-1999, 1999-2003, 2004-2009, 2010-2014, and 2015-2019. Each multi-date composite is visually checked, then a debris-ice transition threshold is chosen automatically with an Otsu routine (Otsu, 1979). By using images stemming from different epochs and through suitable selection (see 3.2 section), the area covered by debris is identified for 68 glaciers, again scattered throughout HMA. All 31 glaciers for which debris-cover is identified on the Hexagon images are also covered by the Landsat data. All glaciers are divided into two sets. The first set (termed S1) is composed of 55 glaciers where debris is identified on Landsat images and 18 glaciers, where the debris is identified from Hexagon imagery. The second set (termed S2), is composed of 13 glaciers where debris is identified on both Landsat and Hexagon images. This division into two sets is done to ensure independence between data used in the calibration and the evaluation of the debris-evolution module.



## 2.3 Mass balance

To calibrate the mass balance module of GloGEMflow, we rely on glacier-specific geodetic volume changes available for
2000-2019 (Hugonnet et al., 2021). These volume changes were obtained from surface elevation changes determined by using
stereo-images from the Advanced Spaceborne Thermal Emission and Reflection Radiometer (ASTER). The volume changes
are converted into mass changes by using a constant density conversion factor of $850 \, \mathrm{kg\,m^{-3}}$ (Huss, 2013). The data set
provides a volume change estimate for all individual glaciers. It covers ≈99.8 % of the regional glacier area (Hugonnet et al.,
2021). For the remaining ≈0.2 %, data from a nearby glacier is chosen by following the same procedure as described in
Compagno et al. (2021). To evaluate the mass balance module, we use independent data from in-situ observations provided by
the World Glacier Monitoring Service for 21 glaciers (WGMS, 2020).

## 2.4 Climate

For forcing GloGEMflow between 1979 and 2020, we use 2m-temperature and precipitation data of the European Centre for
Medium-Range Weather Forecasts Interim re-analysis (ERA-5) (Hersbach et al., 2019). For the future (2020-2100), we use 53
global circulation model (GCM) members of CMIP6 (Eyring et al., 2016), divided into 5 SSPs (5 GCM members for SSP119
and 12 GCM members for all other SSPs). Both data sets have a monthly resolution. To ensure consistency between past and
future, a de-biasing procedure is applied that adjusts the GCMs to the ERA-5 data set (see Huss and Hock, 2015, for details).
This procedure applies a set of additive and multiplicative corrections to adjust the long-term mean difference and the short-
term variability of the coarse-resolution GCMs (with a horizontal resolution of about $100 \, \mathrm{km}$) to the high-resolution ERA-5
data (with a horizontal resolution of about $30 \, \mathrm{km}$).

## 3 Methods

GloGEMflow is a combined mass-balance and ice-flow model, extended with a new component for debris-cover evolution for
this study. The general workflow of this study are illustrated in Fig. 2. In the following sections, the three modules (i.e. the ones
dealing with mass balance, ice flow, and debris cover) are presented.

### 3.1 Mass balance

Accumulation is computed by summing solid precipitation, which is determined by applying a local temperature threshold of
1.5°C (with a linear transition between liquid and solid precipitation in the 0.5 and 2.5 °C range) to the ERA5 grid cell closest
to the glacier. To account for the precipitation increase with elevation, we apply a lapse rate of $0.015 \, \% \, \mathrm{m^{-1}}$ (for consistency,
same as Huss and Hock, 2015). For glaciers with an elevation range over $1000 \, \mathrm{m}$, precipitation is reduced in the uppermost
quarter with an exponential function to account for reduced moisture content in the air and stronger wind erosion (see Huss
and Hock, 2015, for details).



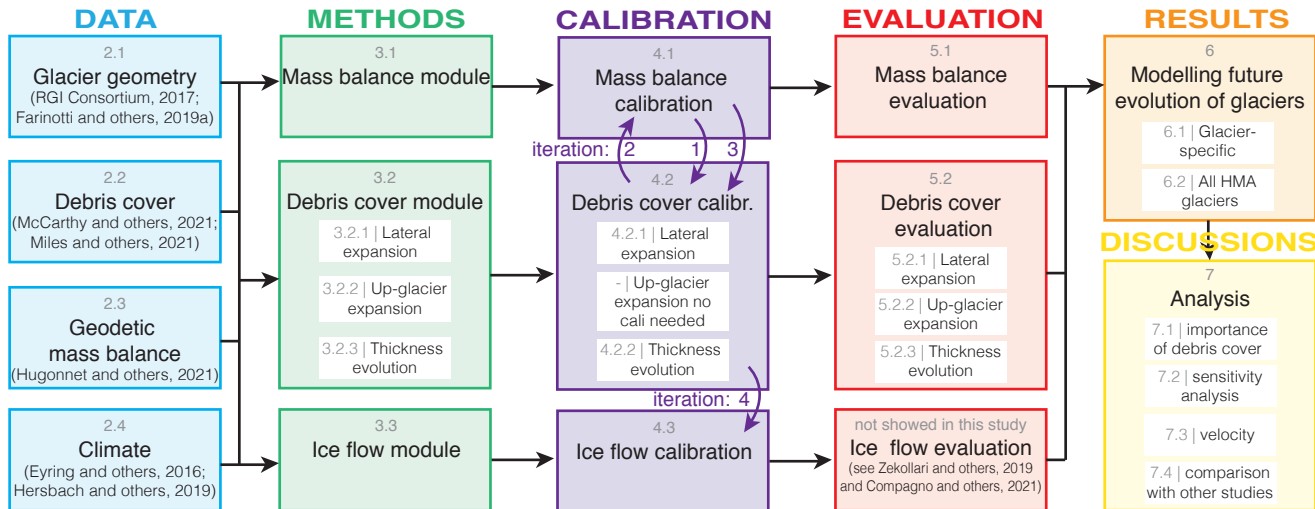

**Figure 2.** Study overview. The grey numbers correspond to the sections of this manuscript. The "METHODS" column also depicts the different modules included in GloGEMflow. The violet arrows show the iterations between the modules during the calibration.

A degree-day model (Hock, 2003) is used to compute ablation. Ice, firn and snow are differentiated by using a different degree day factors (DDF), with a ratio between $DDF_{ice}$ and $DDF_{snow}$ of 2.0, and a ratio between $DDF_{ice}$ and $DDF_{firn}$ of 1.5. The air temperature lapse rate, used to determine temperature for each elevation band of the glacier surface, is computed from

temperature fields at distinct geopotential heights provided by the ERA-5 data set (see Compagno et al., 2021, for more details).

For debris-covered ice, melt enhancement and reduction due to thin and thick debris cover, respectively, is accounted for. This is done by applying a glacier-specific Østrem curve (see Section 2.2) that relates ablation ($a$) under debris to debris thickness ($h$) using Equation 1, while the standard GloGEMflow-calculated ablation (without debris) is used for $h = 0$. In GloGEMflow, the relation between ablation and debris cover is applied to each elevation $z$ and each time step $t$, and can be

expressed as:

$$
\begin{cases}
a_{z,t}^{\text{debris}} = a_{z,t} \cdot g & \text{if } g < 1.65 \\
a_{z,t}^{\text{debris}} = a_{z,t} \cdot 1.65 & \text{if } g > 1.65
\end{cases}
\tag{2}
$$

where $a_{z,t}^{debris}$ (m w. e. a$^{-1}$) is ablation of debris-covered ice, $a_{z,t}$ (m w. e. a$^{-1}$) is ablation of bare ice at elevation $z$ and time $t$. The factor $g$ depends on debris-cover thickness $h_{z,t}$ (m) and the glacier-specific parameter $k_{\text{debris}}$ (see Eq. 1), used by McCarthy et al. (sub.) to fit the Østrem curve. $g$ can be expressed as follows:

$$
g = \begin{cases}
\frac{(k_{\text{debris}} + h_{\text{crit}})}{h_{z,t} + k_{\text{debris}}}, & \text{if } h_{z,t} > h_{eff} \\
\frac{(k_{\text{debris}} + h_{\text{crit}})}{h_{\text{eff}} + k_{\text{debris}}} \cdot \frac{h_{z,t}}{h_{\text{eff}}} + \frac{h_{\text{eff}} - h_{z,t}}{h_{\text{eff}}}, & \text{if } h_{z,t} < h_{eff}
\end{cases}
\tag{3}
$$





where $h_{crit}$ is the critical debris thickness (m), i.e. the debris thickness for which ice melt beneath debris is identical to the melt of bare ice (Reznichenko et al., 2010), and $h_{eff}$ is the debris thickness for which the enhancement of melt is maximal. Here, we use $h_{crit} = 0.036$ m and $h_{eff} = 0.016$ m, which are the means for $h_{crit}$ and $h_{eff}$ as determined from observations by 11 local studies across HMA (Khan, 1989; Mattson and Gardner, 1989; Kayastha et al., 2000; Tangborn and Rana, 2000; Mihalcea et al., 2006; Hagg et al., 2008; Wei et al., 2010; Dobhal et al., 2013; Juen et al., 2014; Sharma et al., 2016; Groos et al., 2018, see Table S1).

In Equation 2, $g$ is constrained to a maximal value of 1.65, which corresponds to the highest observed melt enhancement factor reported in the 11 local sudies (see Table S1). For glacier-specific Østrem curves and examples of $h_{crit}$ and $h_{eff}$, see Fig. S1.

## 3.2 Debris-cover evolution

The evolution of debris-cover extent and thickness is modelled for each glacier with an area >2 km$^2$ and that was identified to be a non-surging glacier in the RGI 6.0. The remaining glaciers are treated as clean-ice glaciers. The evolution of debris-cover extent and thickness is parametrized by accounting for three main processes: (1) the lateral expansion of debris cover within individual elevation bands, which is meant to mimic the observed lateral expansion of medial moraines and debris patches; (2) the debris up-glacier expansion, which describes the progressive appearance of debris at higher elevation when the ELA rises; and (3) debris thickness evolution, which accounts for the progressive accumulation of debris on the surface due to insufficient export by ice flow (see Fig. 3). Note that ponds and ice cliffs, known to influence the surface mass balance of debris-covered glaciers as well (Ragettli et al., 2016; Miles et al., 2018; Rounce et al., 2018), are implicitly accounted for during the Østrem curve fitting procedure (see section 2.2), since their effect is already accounted for in the mass balance data (Miles et al., 2021). We do not model ponds and ice cliffs explicitly because (1) of the lack of detailed information that would be needed for accurate calibration and evaluation at the regional scale, and (2) their long-term and future evolution is uncertain (Narama et al., 2017; Watson et al., 2017; Chand and Watanabe, 2019; Mölg et al., 2020), and requires small-scale, specific process models to be captured (e.g. Buri et al., 2021; Kneib et al., 2021).

### 3.2.1 Lateral expansion of debris cover

The lateral expansion of an already existing debris layer (e.g. medial and ice-marginal moraines, as well as isolated debris patches) is linked to the local mass balance (Stokes et al., 2007; Bhambri et al., 2011; Bolch et al., 2011; Shukla and Qadir, 2016; Tielidze et al., 2020). We describe the process of lateral expansion on a yearly time step by

$$\gamma_{z,t} = \gamma_{z,t-1} + abs(b_{z,t}) \cdot \overline{B_{(t-9,t)}} \cdot (-1) \cdot \gamma_{z,t-1} \cdot c_{\text{lateral}}, \tag{4}$$

where $\gamma_{z,t}$ is the fraction of debris cover in elevation band $z$ at time $t$, $b_{z,t}$ is the mass balance at elevation $z$ at time $t$, and $\overline{B_{(t-9,t)}}$ is the 10-year moving average of the glacier-wide mass balance, evaluated between years $t-9$ and $t$. $c_{\text{lateral}}$ is a debris-cover extension parameter, and is calibrated to minimize the difference between observed and computed lateral expansion of



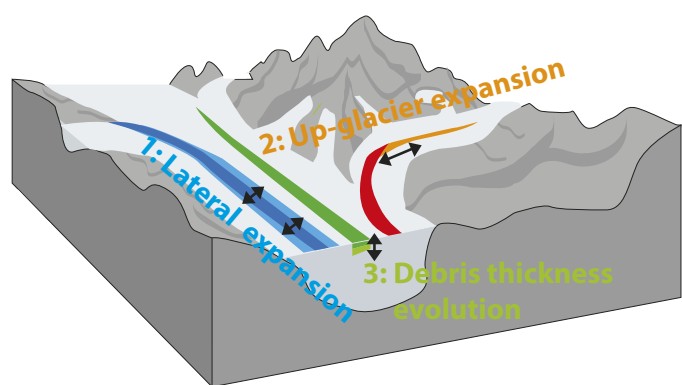

**Figure 3.** Sketch showing the three main processes (parametrizations) captured by the debris-cover extent and thickness evolution module.

debris (see section 4.2). The first term of Eq. 4 accounts for pre-existing debris cover, while the second term describes the rate of debris expansion. The latter is proportional to the local mass balance $abs(b_{z,t})$, which is generally negative where debris cover is present, thus accounting for the expected increase in lateral debris for regions with higher melt rates (Jouvet et al., 2011; Stokes et al., 2007; Bhambri et al., 2011; Bolch et al., 2011; Shukla and Qadir, 2016; Tielidze et al., 2020).

We also consider debris expansion to be inversely proportional to the 10-year moving average of glacier-wide mass balance $\overline{B_{(t-9,t)}}$. By doing so, we parametrize ice-dynamical processes: in the case of negative long-term mass balance, the debris-cover fraction increases, resulting in an accumulation of debris; in the case of positive long-term mass balance, the debris fraction decreases, mimicking debris evacuation by ice flow (Anderson and Anderson, 2016; Ferguson and Vieli, 2021); and

200 in case of neutral long-term mass balance, the debris fraction remains stable. Such a neutral up to positive evolution has for example been observed in the Karakoram over the last 40 years (Herreid et al., 2015). Our implementation also accounts for the observation that in elevations with limited debris, relative expansion is slower compared to elevations with a high debris fraction. Small debris fractions are often associated with small moraines or isolated debris patches, indicative for relatively limited debris concentration in the ice. Areas with already abundant debris cover may grow faster, due to more important

debris supply from melt-out or from the surroundings (Anderson, 2000).

### 3.2.2 Up-glacier expansion of debris cover

For glaciers with negative mass balances, debris cover has been observed to progressively expand up-glacier (Stokes et al., 2007). This is related to the rise of the ELA and the melt-out of debris transported by ice flow into areas that transit from the accumulation to the ablation zone (Anderson, 2000). As Eq. 4 does not permit simulating the expansion of debris to new

elevation bands, we parameterize this process as:

$$\gamma_{z,t} = \frac{\gamma_{z-1,t} + \gamma_{z+1,t}}{2}, \quad \text{if } \gamma_{z,t-1} = 0. \tag{5}$$





This process is discretized with elevation bands of 10 m. The amount of elevation bands without debris at time $t-1$ that can gain debris from a nearby elevation band at time $t$ (we use yearly time steps) applying Eq. 5, is proportional to the rise of the ELA over the last ten years, determined using linear regression ($\text{ELA}_{(t-9,t)}$). In other words, the up-glacier expansion of debris cover rises with the ELA rise. The thickness of new debris is arbitrarily set to 1 cm. If the slope of the above linear regression is zero or negative, Equation 5 is not applied. With the above process, debris migrates towards higher elevations at the same rate as the ELA rises. If the ELA doesn't rise, the maximal elevation at which debris is encountered will remain stable, or decrease if the glacier mass balance is positive, e.g. due to negative lateral expansion of debris cover (see section 3.2.1). The procedure has no calibration parameters, and is evaluated in section 5.2.2.

### 3.2.3 Debris thickness evolution

As for the lateral expansion of debris, the evolution of debris thickness is linked to internal debris concentration and glacier mass balance (e.g. Gibson et al., 2017; Mölg et al., 2019; Verhaegen et al., 2020). Additionally, external drivers such as rock avalanches may locally control the debris thickness (Shugar and Clague, 2011; Dunning et al., 2015; Berthier and Brun, 2019). We parametrize the change in local debris thickness based on an approach that is structurally similar to the one used for lateral debris expansion:

$$h_{z,t} = h_{z,t-1} + abs(b_{z,t}) \cdot \overline{B_{(t-9,t)}} \cdot (-1) \cdot \overline{h_0} \cdot c_{thickening}, \quad \text{if } \gamma_{z,t-1} = 0 \tag{6}$$

where $h_{z,t}$ is the debris thickness for elevation $z$ and time $t$. $c_{thickening}$ is a calibration parameter for the debris-cover thickness evolution, constrained based on observations (see section 4.2). As for lateral debris expansion, the local mass balance $b_{z,t}$ relates linearly to debris thickness change. Higher melt rates will lead to faster debris thickening, thus implicitly assuming that debris concentrations within the ice are homogeneous. The long-term glacier-wide mass balance $\overline{B_{(t-9,t)}}$ mimics ice-dynamical processes. It leads to constant debris thickness for steady-state conditions ($\overline{B_{(t-9,t)}} = 0$), and to decreasing local debris thickness for consistent mass balances, thus mimicking the evacuation of debris with enhanced flow. This is in line with the few direct observations that are available (e.g. Gibson et al., 2017; Verhaegen et al., 2020). $\overline{h_0}$ is the mean debris thickness of the glacier at the inventory year. It parametrizes the effect that glaciers with a low mean debris thickness will thicken slower compared to glaciers with a high mean debris thickness. This is motivated by the assumption that glaciers with thick debris are likely to have a higher englacial debris concentration, indicative for high debris supplies from the surroundings.

### 3.3 Ice-flow

For modelling glacier's geometry evolution, ice flow is explicitly accounted for based on the shallow-ice approximation and the continuity equation (see Zekollari et al., 2019, for details). This is done for all glaciers with an area >2 km$^2$, the ice flow being controlled by a deformation-sliding factor that accounts for both internal ice deformation and basal sliding. This deformation-sliding factor is calibrated for each glacier specifically (see section 4.3). For all glaciers with an area <2 km$^2$,





glacier evolution is modelled with an elevation-dependent parameterisation, which was shown to be in good agreement with results from higher-order ice-flow models (Huss et al., 2010).

## 4 Model calibration

The interaction between the modules (especially of the calibration) and the general workflow of this study are illustrated in Fig. 2. First, the mass balance module is calibrated (see section 4.1), followed by the calibration of the debris-cover evolution module (see section 4.2). This procedure is iterated twice since debris evolution feeds back to mass balance. Finally, the ice flow module is calibrated (see section 4.3), and the three modules are evaluated independently (section 5).

### 4.1 Mass balance

A glacier-specific, three-step calibration procedure is used to account for the sensitivity of each glacier to the local climate (Huss and Hock, 2015, as used in). The goal is to match the glacier-specific mass balance between 2000 and 2019 provided by Hugonnet et al. (2021). The accepted misfit is of 0.01 m w.e.a$^{-1}$. First, the precipitation given by the forcing data set is adjusted with a multiplicative enhancement factor that is allowed to vary between 0.6 and 2.0. Second, the degree-day factors are varied in a range of between 1.75 and 4.5 mm d$^{-1}$ K d$^{-1}$ for DDF$_{snow}$, and DDF$_{ice}$ prescribed to always relate with a factor

2 to DDF$_{snow}$. Third, the local air temperature is adjusted. The steps are applied sequentially, meaning that the calibration is considered to be completed as soon as the observations are matched within the tolerated misfit (see Huss and Hock, 2015, for more details). Steps 2 and 3 may thus not be applied in all cases. Indeed, 44 % of the glaciers found an agreement in the first step, 30 % in the second and 26 % in the third step.

In order to investigate the importance of the new debris-cover module when projecting future glacier evolution (through
the paper, this approach is termed "explicitly" accounting for debris cover), we establish a second glacier-specific parameter set where all parameterizations related to debris cover are disabled (through the paper, this approach is termed "implicilty" accounting for debris cover). In this case all glaciers are regarded as clean-ice glaciers. As we use observed geodetic mass changes for calibration, however, this parameter set accounts for the effect of debris cover implicitly. We re-calibrate GloGEMflow only by adjusting the DDF (step 2) but by keeping unaltered the model parameters determined in step 1 (and potentially step 3). This
strategy thus preserves the glacier-specific climate conditions (precipitation totals and temperature) but adjusts the glaciers' temperature-sensitivity for snow and ice melt in order to reproduce the observed mass change even without directly accounting for the melt-reduction process of supraglacial debris coverage.

### 4.2 Debris-cover evolution

#### 4.2.1 Calibrating lateral debris expansion

To determine $c_{lateral}$, we use the debris-cover observations of the Landsat scenes of set S1 (see section 2 and e.g. Fig. 4a). First, the evolution of the debris' lateral expansion is calculated for each glacier and each elevation band (e.g. Fig. 4b). Then,





calibration is performed by comparing the lateral expansion of debris as observed and as modelled using different $c_{\mathrm{lateral}}$ factors (ranging from 0 to 5). More specifically, we take the debris extent detected on each Landsat scene as the initial condition, and we simulate each glacier independently for each $c_{\mathrm{lateral}}$. Finally, we calculate the root-mean-square error (RMSE) between

modelled and observed lateral debris expansion over the period captured by our data set and for each of the 55 glaciers. For each glacier, we select the $c_{\mathrm{lateral}}$ which results in the lowest RMSE (see Fig. 4c). The mean of the selected $c_{\mathrm{lateral}}$ is $c_{\mathrm{lateral}} = 2.0$, while the 0.25 and 0.75 quantiles are $c_{\mathrm{lateral}} = 0.4$ and $c_{\mathrm{lateral}} = 4.2$, respectively. The mean value is used for all further modelling, whilst the result's sensitivity to the spread in $c_{\mathrm{lateral}}$ is analyzed in section 5.2.1.

### 4.2.2   Calibrating debris thickness evolution

To determine $c_{thickening}$, a three-step procedure is used. In a first step, we map where debris cover appeared for the first time between $\approx 1974$ (Hexagon satellite imagery) and $\approx 1989$ (oldest Landsat satellite image). This is done for 12 glaciers with Hexagon satellite observations in the S1 set (out of the 18 glaciers in total of set S1) within three subregions (Central Himalaya, East Himalaya and West Tien Shan). The six remaining glaciers are not used because a clear signal of debris formation is lacking between $\approx 1974$ and $\approx 1989$. In a second step, we extract the debris thickness at the locations used in

McCarthy et al. (sub.)'s debris thickness data set. Recall that the latter data set represents the debris condition for 2000-2016. Combined, this information provides us with an estimate for the mean debris thickening rate between $\approx 1981$ (mean between 1974 and 1989) and $\approx 2008$ (mean between 2000 and 2016). In a third step, we compute the difference between observed and modelled debris-thickening rate for each glacier. To do so, the 12 selected glaciers are modelled with $c_{thickening}$ values ranging between 0 and 2 (Fig. 5), and the value minimizing the difference to observations is chosen. We find $c_{thickening} = 1.0$.

This value and its sensitivity are evaluated in Section 5.2.2.

### 4.3   Ice flow

The ice flow module is initialized and calibrated by generating a glacier-specific steady state for a specified point in time in the past. The exact timing of this point in time depends on both climate and glacier response time (see Compagno et al., 2021, for more details). Starting from this steady state (on average between 1981±2 and 1993 ±3 in this study), the glacier is transiently

modelled up to the glacier inventory date by using ERA-5 climate data. To ensure that the so-modelled glacier volume and length are consistent with the available observations, the procedure is repeated by iteratively changing two parameters: the deformation-sliding factor and a mass-balance bias applied during the generation of the steady state (see Zekollari et al., 2019, for more details). Since the entire procedure is performed before the glacier-specific inventory year, debris cover is considered to be static (as given by the observations). Once calibrated, the model is forced by ERA5-climate data (until 2020), and GCM

output data to simulate the future glacier evolution (2020 until 2100).



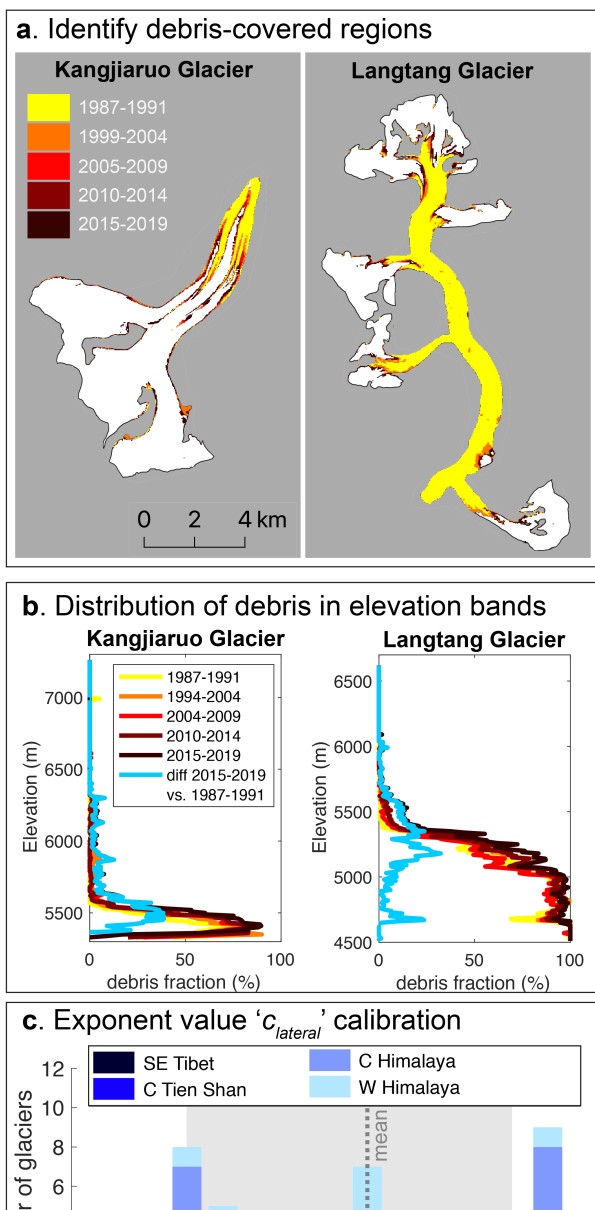

**Figure 4.** (**a**) Evolution of the debris-covered area of Kangjiaruo Glacier and Langtang Glacier as inferred from five Landsat scenes. (**b**) Same as (**a**), but divided into 10-meter elevation bands. The blue line shows the lateral expansion of the debris cover as observed between the oldest (1987-1991) and the newest (2015-2019) Landsat scene. (**c**) Distribution of $c_{\text{lateral}}$ resulting in lowest misfit between observed and modelled derbis fraction evolution (given per number of glaciers). The grey dashed line shows the mean value while the grey rectangle shows values within the 0.25 and 0.75 quartiles.





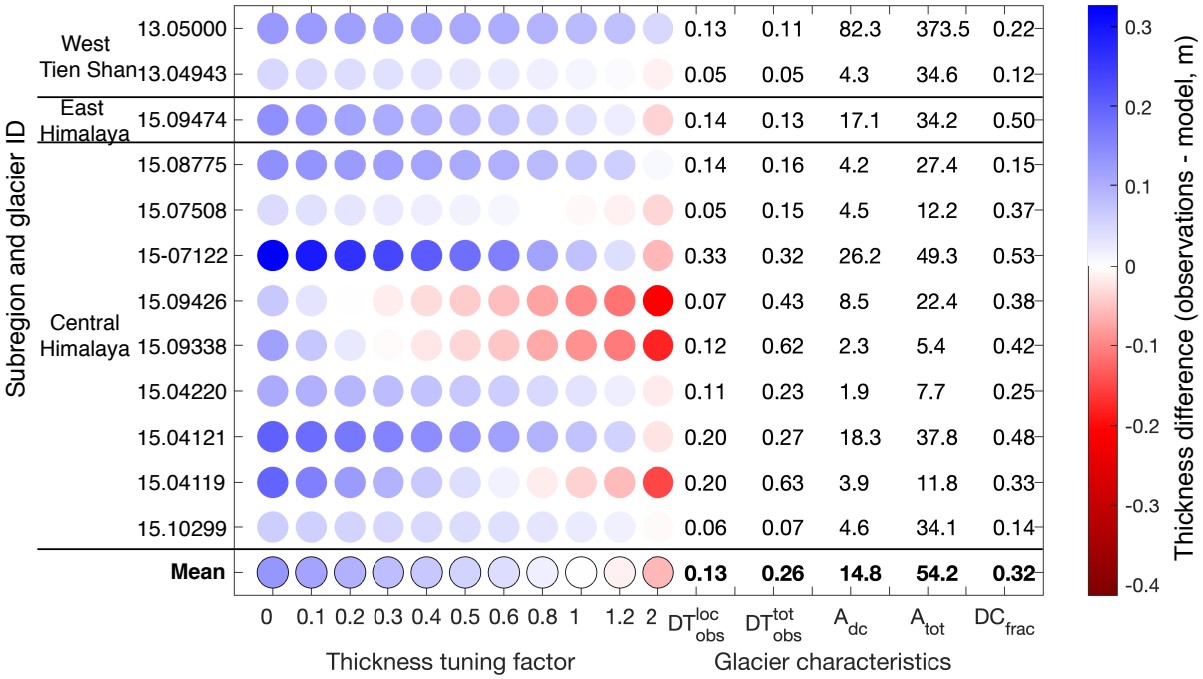

**Figure 5.** Difference between observed and modelled debris thickness for the period $\approx$ 1981–2008 (circles). The numbers in the right-hand part of the figure are glacier specific . $DT_{\mathrm{obs}}^{\mathrm{loc}}$ is the observed mean debris thickness (m) for locations in which the debris was formed between $\approx$ 1974 and $\approx$ 1989; $DT_{obs}^{tot}$ is the mean debris thickness (m) of the debris-covered part at inventory date; $A_{dc}$ is the debris-covered area (km$^2$); $A_{\mathrm{tot}}$ is the area of the whole glacier (km$^2$); and $DC_{\mathrm{frac}}$ is the lateral expansion of the debris cover, i.e. $DC_{\mathrm{frac}} = A_{\mathrm{dc}} / A_{\mathrm{dc}}$ (%).

# 5 Model evaluation

## 5.1 Mass balance

To evaluate the performance of the mass-balance module, we compare the modelled mass balances for 21 glaciers in HMA against observations provided by the World Glacier Monitoring Service (WGMS, 2020). For glacier-wide annual mass balance,

the bias (measured − modelled) is –0.24 m w.e. a$^{-1}$ and the RMSE is 0.55 m w.e. a$^{-1}$ (see Fig. S2). Observations aggregated to elevation bands show a bias of $-0.38$ m w.e. a$^{-1}$ and a RMSE of 0.77 m w.e. a$^{-1}$. For glacier-wide winter balance, the bias is 0.23 m w.e. a$^{-1}$ and the RMSE is 0.41 m w.e. a$^{-1}$ (see Fig. S3). These results are satisfactory in comparison to other regional-scale modelling studies (e.g. Marzeion et al., 2012; Huss and Hock, 2015; Radič and Hock, 2014).

## 5.2 Debris evolution

In this section, the three parametrizations included in our debris-evolution module (see Fig. 3) are evaluated against independent data sets.



### 5.2.1 Evaluating lateral expansion of debris

To evaluate the parametrization for lateral debris expansion, 18 glaciers (S1 with Hexagon satellite observations) within three sub-regions (Central Himalaya, West Himalaya and West Tien Shan) are simulated from 1974 to 2020. The model is initialized with debris extents extracted from Hexagon satellite images of ≈1974 (see section 2). The modelled debris-area evolution is evaluated between 1989 and 2017 against the time series of debris extents obtained from Landsat (three to five observations per glacier). We calculate the misfit between modelled and observed debris-cover fraction for each elevation band. Note that this is the same procedure as used for calibration (see Section 4.2), with the difference that for model initialization we use debris extents from Hexagon imagery rather than Landsat (Fig. 6). The mean misfit of debris fraction obtained by this procedure is 0.7 % (Fig. 6a, blue histogram), with 42 of the 78 evaluations indicating a misfit $< \pm\, 2\,\%$ (Fig. 6a/b). A good model performance is also shown by analyzing the glacier-specific lateral expansion of debris (Fig. 6c/d).

The performance of our parametrization for lateral debris expansion can also be evaluated by disabling this process in the model. We do so by initializing the model as above but by prescribing a constant debris cover, taken to be the one of ≈1974. In this case, the mean misfit between observed and modelled debris fraction is 4.4 % (Fig. 6a, red histogram), i.e. substantially higher than for the case in which the debris area evolution is included. The experiment thus shows the importance of accounting for debris expansion when modelling long-term glacier evolution.

### 5.2.2 Evaluating debris up-glacier expansion

To evaluate the parametrization for the up-glacier debris expansion, we use the same experiment setup as above (Section 5.2.1). We initialize the model in 1974 and force it until 2020 but now focus on the transition zone between debris-covered and bare-ice surfaces. For each glacier with observed debris extents from Hexagon and Landsat (of set S1), we extract from the Landsat scenes the highest elevation bands that have a debris-covered fraction of $\geq$10, 20, 30, 40 and 50 %. We perform the same extraction procedure for the modelling results. Then we compute for each glacier and each Landsat scene the elevation misfit between observations and modelling results (Fig. 7). The mean misfit is of $+19$ m.

Again, an alternative way for evaluating our approach is to turn off the up-glacier expansion module, thus prescribing a temporally constant debris cover, set to be the one inferred for ≈1974. For this case without up-glacier debris expansion, we re-compute the misfit between observed and modelled elevation with a debris fraction $\geq$10, 20, 30, 40 and 50 %. This results in a misfit of $+55$ m, i.e. almost three times larger compared to when the up-glacier debris expansion module is activated, indicating the importance of taking up-glacier debris expansion into account as well.

### 5.2.3 Evaluating debris thickness evolution

To evaluate the debris thickness evolution parametrization, we compare model results against the evaluation data set S2, i.e. 13 glaciers distributed in four regions (East Himalaya, Karakoram, West Tien Shan and Inner Tibet). By setting $c_{thickening} = 1.0$, the mean misfit between observed and modelled debris-thickening rate is 0.10 m. For $c_{thickening} = 0.0$ (i.e no debris thickness evolution) it is 0.19 m (see Fig. S4). The model, thus, performs better when the debris thickness evolution parametrization is



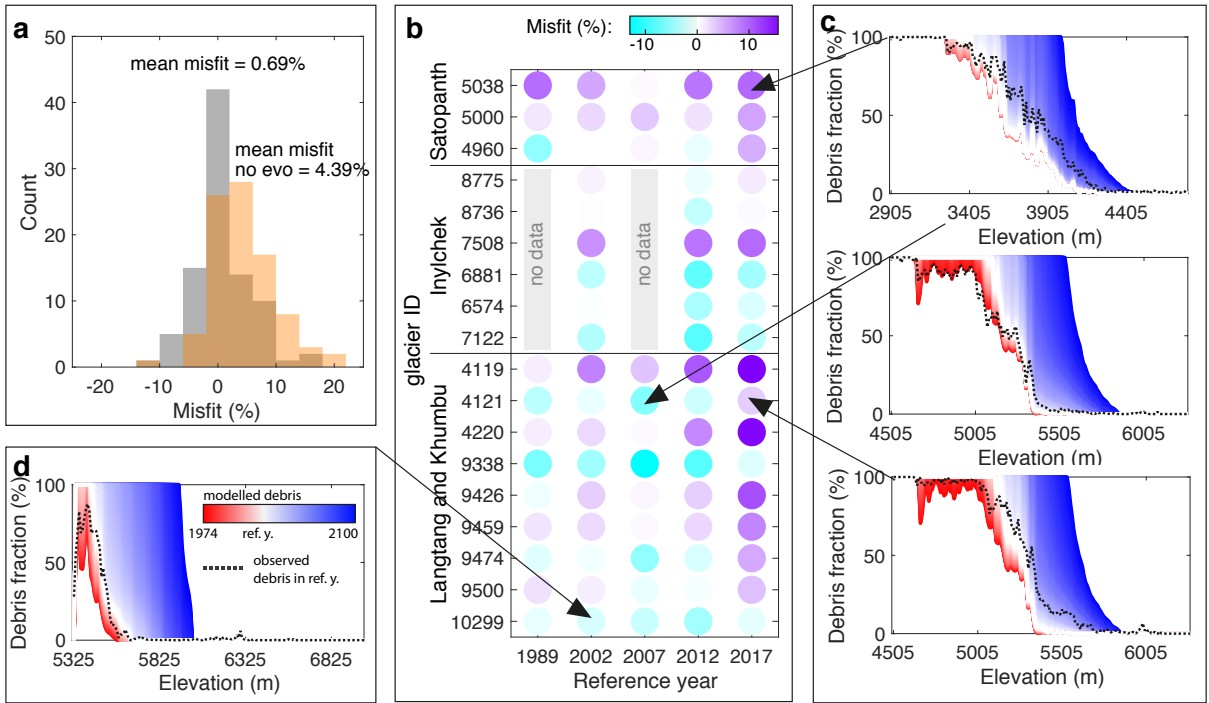

**Figure 6.** (**a**) Histograms of the misfit between observed and modelled lateral debris expansion when using $c_{lateral} = 2$ (grey) and when deactivating the module ($c_{lateral} = 0$, orange). (**b**) Same as (**a**) but divided into glaciers and distinguishing between different reference years (corresponding to the Landsat scenes). The colour of each circle represent the misfit (%). (**c** and **d**) modelled debris-area evolution (red-white-blue shades) and debris-area evolution as observed in the Landsat scenes (black dashed line) for four selected glaciers (indicated by arrows).

activated. Taken together, this evaluation and the calibration results (section 4.2.2) indicate high glacier-to-glacier variance of

345 $c_{thickening}$, but also show that the proposed parametrization is rather insensitive to the weakly constrained value of $c_{thickening}$ (see also section 7).

# 6 Results

## 6.1 Glacier-specific simulations

In order to illustrate the detailed model results at the scale of an individual glacier, we focus on the well-investigated Langtang

Glacier, Central Himalaya. A similar illustration for Baltoro Glacier (Karakoram) and Inylcheck Glacier (Western Tien Shan), which showed similar patterns like Langtang Glacier, is given in Supplementary Figures S5 and S6, respectively.

  Figure 8a shows a profile view of the glacier and debris-cover evolution of Langtang Glacier according to SSP245. Under this scenario, Langtang Glacier would lose 43 % of its 2020 ice volume by 2050, despite of the terminus retreating by less than



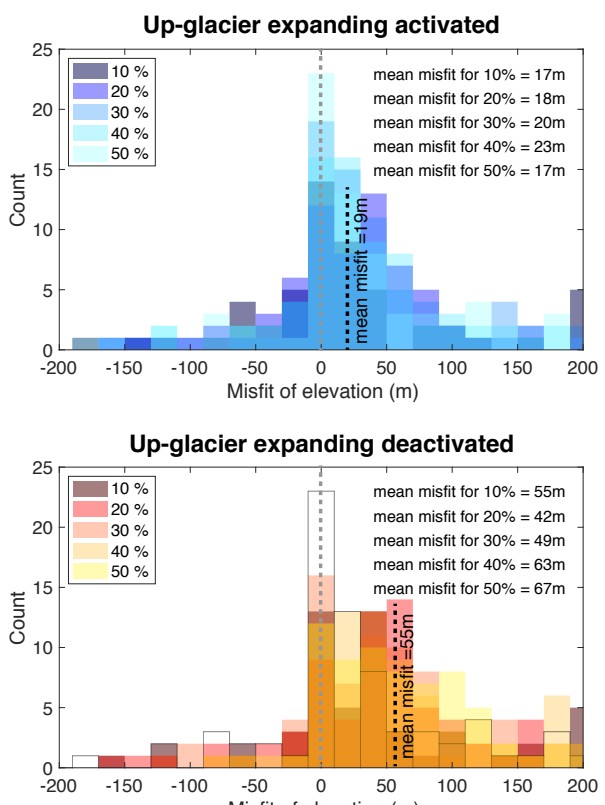

**Figure 7.** Misfit between observed and modelled highest elevation with a lateral debris expansion ≥10, 20, 30, 40 and 50 %. The debris area evolution module is activated in panel **a** and deactivated in panel **b**.

300 m (i.e. only 2 % of its 2020 length). The limited retreat can be attributed to the 0.5-1 m thick insulating debris cover present
on the entire glacier tongue, which reduces Langtang Glaciers's ice melt by a factor of about three compared to the hypothetical situation with no debris. The approximately linear dependence between surface mass balance and elevation, which is typical for clean-ice glaciers, is suppressed for Langtang Glacier since debris cover is thicker at lower elevation than it is for higher ones (Bisset et al., 2020; Miles et al., 2021). This leads to a nearly homogeneous downwasting of the glacier (e.g. Pellicciotti et al., 2015; Ragettli et al., 2016) rather than to a retreat of the terminus (e.g. Benn et al., 2012). This causality is confirmed
when the evolution of Langtang Glacier is re-computed using the same climatic conditions but when re-calibrating the model parameters to match the observed volume changes without activating the debris-cover module (Fig. 8b). In this case, the glacier would lose 45 % of its 2020 ice volume by 2050 (i.e. very similar as with explicit debris modelling), but would retreat by about 2700 m (i.e. 20 % of its 2020 length). The latter is 10 times more than when including the effect of supraglacial debris.

Figures 8a and c show a spatial representation of the debris-cover evolution according to the three implemented parametriza-
tions. At 5500 m a.s.l for instance, the fraction of debris increased by 87 % between 2025 and 2100. This result is driven by

**Figure 8.** (**a**) Modelled evolution of Langtang Glacier when debris is explicitly accounted for. The results refer to SSP245. Note that the debris thickness (grey) is exaggerated by a factor of 500 for visibility. The three parametrizations included in the debris-cover module (cf. section 3.2 and Figure 3) are indicated by the circled, colored numbers, and described in the text. (**b**) Same as (**a**), but accounting for debris implicitly, i.e. glacier evolution is not modelled with the new debris module, but by re-calibrating some of the model parameters to match observed long-term mass balance (see section 4.1 for details). (**c**) Model results extrapolated to 2D (see Supplementary Material for the extrapolation method). For every SSP, the evolution of (**d**) debris-cover fraction, (**e/f**) glacier volume with explicit/implicit debris-cover modelling, (**g**) debris thickness, and (**h/i**) glacier area with explicit/implicit modelling is shown. The shaded ranges represent one standard deviation of all climate model members included in a given SSP.





the projected lateral debris expansion. As a result of the projected up-glacier migration, instead, the maximum elevation with supraglacial debris increases by 280 m for the same time period. Finally, the local debris thickness at 5500 m a.s.l is projected to increase by 0.23 m over the period 2025-2100.

The presence of supraglacial debris alters glacier mass balance, hence influencing the debris-cover evolution itself. With
higher mass loss, both debris-covered area and debris thickness increase, thus reducing ice melt. Nevertheless, higher mass loss also leads to glacier retreat and downwasting of the debris-covered glacier tongue, thus reducing debris-cover extent due to glacier area loss. Therefore, a competition between debris increase and reduction arises after 2060. For Langtang Glacier, model simulations show that the fraction of debris-covered area is expected to increase until 2060 reaching a maximum of 55±2 % (mean and standard deviation of all model members considering SSP245) relative to the remaining glacier area. After
reaching this peak debris fraction, a fast decline is modelled due to disintegration of the debris-covered tongue. Depending on the emission scenario, the debris-covered fraction reaches between 28±6 % (SSP119) and 18±5 % (SSP585) by 2100 (Fig. 8f). Note that the fraction refers to the evolving glacier geometry, and not to the geometry at inventory date. By turning off the debris-evolution module (expansion and thickening) and by using presently observed debris extent and thickness instead (grey line in Fig. 8f), the debris fraction would continuously decrease, reaching between 14 ±7% (SSP119) and 1±1 % in 2100
(SSP585).

Compared to 2020, the mean debris thickness of Langtang Glacier is expected to increase by 35±5 % and reach its maximum in 2065 (SSP245). The variation between individual SSPs is relatively small. Different SSPs give rise to different thickness evolution trajectories however, reaching between −7±10 % (SSP119) and −75±19 % (SSP585) of the 2020 debris thickness by 2100. This counter-intuitive decrease in average debris thickness can be explained by the evolution of both debris extent
as well as glacier geometry. Indeed, the expansion of thin debris to higher areas and the loss of presently thick debris on the downwasting tongue results in a mean debris thickness decrease (see also section 7). When neglecting the evolution of debris extent and thickness (i.e. the change of debris thickness is due only due to glacier geometry change), the mean debris thickness would decrease by between 30±18 % (SSP119) and 80±35 % (SSP585) by 2100. Together with the debris-fraction evolution, this demonstrates that it is relevant to account for transient debris-cover changes in process-based models, at least for low- to
medium emission scenarios (Fig. 8d).

By 2100, and when explicitly modelling debris-cover changes, Langtang Glacier is projected to lose between 69±14 % (SSP119) and 98±2 % (SSP585) of its 2020 ice volume (Fig. 8e). If debris cover is implicitly taken into account, very similar results are obtained, with simulated 2100 area and volume loss only differring by between 1 and 6 %, depending on the SSP (Fig. 8h/i). This indicates that constraining the model to past glacier mass loss yields similar results, even when considering
Langtang Glacier to be a clean-ice glacier. In that case, however, surface mass balance gradients would be erroneous (see e.g Fig. S7), with consequences for the geometry evolution, runoff and surface-elevation feedbacks.

## 6.2 Regional glacier evolution

The area-averaged debris-cover fraction of all glaciers in HMA is expected to be between 14% and 24% by 2100 compared to the 12-13% observed today (Fig. 9a; note that these numbers refer to non-surging glaciers with an area >2 km$^2$, and that the



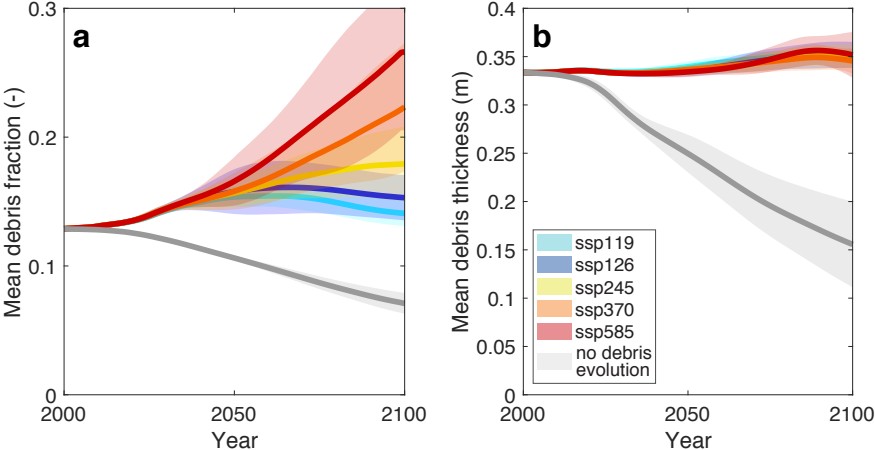

**Figure 9.** Evolution of (**a**) debris-covered fraction and (**b**) debris thickness for all HMA glaciers and as an average for the respective SSP. The shaded bands represent one standard deviation of all climate model members. The grey line indicates the case in which the debris-evolution module is disabled and only today's debris cover is used in the modelling.

debris-cover fractions are computed for the transiently evolving glacier areas). Generally, the debris-cover fraction is projected to be higher for higher emission scenarios (i.e. scenarios implying higher air temperature increase). The expected increase in debris-cover fraction is due to both lateral and up-glacier expansion. Without accounting for debris-cover evolution, the debris-cover fraction, however, is projected to be between $8\pm1\%$ (SSP119) and $6\pm1\%$ (SSP585) relative to today's value. This highlights the importance of accounting for dynamic debris-cover evolution in process-based studies. Our results also show

that under low-emission scenarios, the competing processes of debris expansion and glacier retreat tend to reach an equilibrium at the end of the century. For high-emission scenarios, instead, debris-cover expansion dominates over glacier retreat.

    Between 2020 and 2100, the area-averaged debris-cover thickness of all glaciers in HMA (again, with area $>2\,\mathrm{km}^2$ and not of surge-type) is expected to slightly increase by about 5%, or 2 cm compared to today, see Figure 9b. Interestingly, a very similar change is found for both low- and high-emission scenarios. Locally, however, much higher debris thickness increase are

found but these are offset at many places by the overall reduction in glacier area. Indeed, the small overall change is explained by (1) the disintegration of glacier tongues, where the debris is generally thickest, (2) lateral debris expansion, which is most efficient at intermediate elevations with relatively thin debris, and (3) up-glacier expansion of debris, which forms new thin debris. If debris evolution is not modelled (i.e. static debris), the modelled overall change in debris thickness would be more pronounced, with an average of between $-33\pm10\%$ (SSP119) and $-66\pm5\%$ (SSP585) for the period 2020-2100.

By 2100, HMA glaciers are expected to lose between $35\pm15\%$ (SSP119) and $80\pm11\%$ (SSP585) of their 2020 ice volume when debris cover is explicitly modelled (Fig. 10a). By modelling debris-cover implicitly (i.e. using the same climatic conditions together with parameters re-calibrated to match observed volume change but without activating the debris-cover module), the simulated mean ice volume loss would be only between 1% (SSP119) and 3% (SSP585) higher (Fig. S8a). The difference

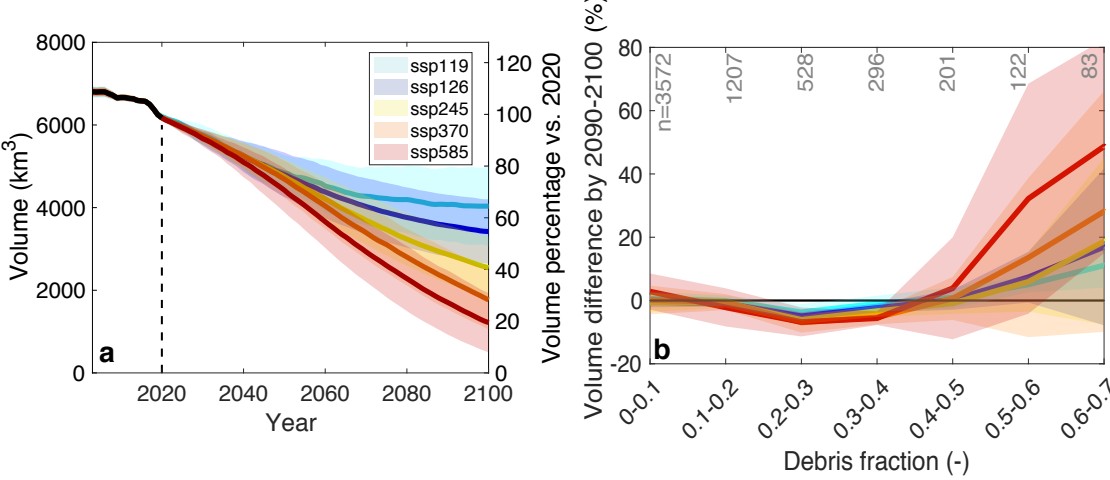

**Figure 10.** (**a**) Evolution of the glacier volume for all glaciers in HMA when explicitly modelling debris-cover changes. Results are aggregated to the five SSPs. (**b**) Difference in volume (mean over 2090-2100) between implicit and explicit debris-cover modelling classified for different debris-cover fractions. The number of modelled glaciers per class is given in grey. The shaded bands represent one standard deviation of all climate model members for a given SSP.

is small because (1) only about 12-13% of the glacier area in HMA is currently debris-covered (Herreid and Pellicciotti, 2020),
(2) the models are constrained to reproduce observed mass change, both when accounting and neglecting the effects of debris cover, and (3) both positive and negative differences can be found at the level of individual glaciers, caused by different debris and geometry evolution of each glacier. By dividing glaciers into classes of debris-cover fraction, the difference between modelling debris cover explicitly or implicitly becomes more evident, reaching up to 30 % of difference in volume for glaciers with a present debris fraction >0.5 (see Fig. 10b). This shows that the difference driven by non-linear feedback of debris cover on
mass balance becomes relevant for glaciers with extensive debris-cover fractions. For differences in both future area evolution and spatially distributed glacier evolution, see Figures S9 and S10.

# 7   Discussion

## 7.1   Importance of accounting for debris-cover evolution

We demonstrated the difference between explicitly and implicitly accounting for the effect of debris cover, as well as the
importance of modelling debris-cover evolution. The goal was to assess whether such processes need to be taken into account when modelling glacier evolution at the local to regional/global scale. The most significant differences emerge for computed glacier length changes, with further differences being found for volume and area evolution, especially for glaciers with high debris-cover fraction and thickness at present (see example of Langtang Glacier; Fig. 8a/b). Moreover, modelled surface mass balance gradients also differ when not explicitly accounting for debris cover (see Fig. S7). Even though for all glaciers the





mass change between 2000 and 2019 is constrained to match the same data (Hugonnet et al., 2021), the spatial mass balance
distribution influences the geometry evolution and, hence, mass turnover and surface flow velocity of glaciers.

Aggregated over all of HMA and considered in terms of glacier volume and area changes, the difference between explicitly
and implicitly modelling debris cover is relatively small. This, however, does not mean that debris can be neglected in climate
impact studies. In fact, accounting for the debris cover explicitly enables the model to correctly represent the driving processes,
rather than compensating the lack of model capabilities through a suitable parameter choice. This is important, especially
when results other than area and volume changes are of interest. Indeed, quantities such as the local mass balance, the glaciers'
ice flow velocity and mass turnover, the glacier's length change or water runoff are only captured correctly when explicitly
accounting for supraglacial debris and its temporal evolution. These quantities, in turn, have to be modelled correctly when
aiming at anticipating other glacier-related processes, such as hazards from ice-dammed or proglacial lakes, or potential slope
instabilities.

Compared to the static representation of supraglacial debris cover that is presently included in some regional to global glacier
models, the expected increase in both debris-cover fraction and local debris thickness will enhance the insulating effects of the
debris cover. Figure 9a/b shows that if the debris area and thickness would not evolve through time, the future debris-cover
fraction and mean debris thickness would significantly decrease. This is related to the loss of frontal area projected in such a
case, since the frontal area typically features the highest concentration of supraglacial debris. However, a significant decrease
in debris cover has neither been observed over the past decades (Stokes et al., 2007; Bhambri et al., 2011; Bolch et al., 2011;
Shukla and Qadir, 2016; Tielidze et al., 2020) nor has it been modelled in glacier-specific studies specifically addressing the
future evolution of debris (Jouvet et al., 2011; Rowan et al., 2015; Kienholz et al., 2017; Verhaegen et al., 2020).

## 7.2 Model sensitivity and uncertainties

Uncertainties in glacier and debris evolution can arise from many factors. In Zekollari et al. (2019) and Compagno et al. (2021),
limited sensitivity to the initial ice thickness distribution, the geometry-initialization method, the model parameters and the
data used for mass balance calibration was found. In this study, additional uncertainties arise from the parameters $c_{\mathrm{lateral}}$ and
$c_{\mathrm{thickening}}$ of the debris-evolution module, i.e. from the lateral debris expansion and thickening parameters. To test model
sensitivity to variations in these factors, we re-compute the future evolution of all glaciers with $c_{\mathrm{lateral}} = 1.0$ and $c_{\mathrm{lateral}} = 3.0$
(compared to the reference value $c_{\mathrm{lateral}} = 2.0$), and with $c_{thickening} = 0.5$ and $c_{thickening} = 1.5$ (reference: $c_{thickening} = 1.0$).
The results indicate that these variations in both factors have little impact on the ice volume loss modelled for 2020-2100, with
a difference of less than $\pm 1\%$. By disabling the debris-evolution module, (i.e. with $c_{\mathrm{lateral}} = 0$ and $c_{thickening} = 0$), the regional
ice volume loss would be about 1 % higher. This small difference is again due to the fact that about 87-88 % of glacier area is
debris-free. However, volume differences for individual large and strongly debris-covered glaciers can be as high as 18% (e.g.
Langtang Glacier 2 %, Baltoro Glacier 8 % and Inylcheck 1 %).



## 7.3 Velocity

To assess the effect of explicitly accounting for debris-cover dynamics on the modelled mass turnover, we compare the computed surface velocities against NASA's MEaSUREs ITS_LIVE surface velocity data set (Gardner et al., 2019). The comparison is performed for all glaciers in HMA with a debris-cover fraction >0.3. For ITS_LIVE, we use the 120 m resolution
composite data covering the period 1985-2018. To account for spatial variations in surface velocity, we compare modelled and observed surface velocities aggregated to 100 m elevation bands at the scale of each individual glacier. We exclude the glaciers' accumulation area since the insufficient contrast in optical images impede feature tracking, thus resulting in higher uncertainties in the ITS_LIVE velocity.

About two thirds of the 3,767 elevation bands investigated with this selection show a velocity that is closer to observations
when the debris cover is explicitly accounted for. The mean modelled velocity is 16 % slower than in the case in which the debris cover is neglected (Figure S11). This indicates that the mass turnover is indeed smaller when debris cover is accounted for, which is consistent with the available ITS_LIVE observations. The smaller ice velocities are in line with findings of more theoretical and process based modelling studies on the dynamics of debris covered glaciers (Anderson and Anderson, 2016; Ferguson and Vieli, 2021)

## 7.4 Comparison with other studies

We compare our HMA-wide results against the ones from the nine global glacier models (Van de Wal and Wild, 2001; Marzeion et al., 2012; Radić and Hock, 2014; Huss and Hock, 2015; Kraaijenbrink et al., 2017; Sakai and Fujita, 2017; Maussion et al., 2019; Shannon et al., 2019; Rounce et al., 2020) that participated in the Glacier Model Intercomparison Project, phase 2 (GlacierMIP2, Marzeion et al., 2020). Since models participating in GlacierMIP2 used CMIP5 GCMs to force the glacier
evolution models, we additionally compare our results to those of Edwards et al. (2021), who used statistical emulation to convert the glacier volume evolution projected by Marzeion et al. (2020) from CMIP5 to CMIP6. The individual GlacierMIP2 models used various methods for modelling both the evolution of glacier geometry and glacier mass balance, as well as various spatial discretizations (refer to Table 1 in, Marzeion et al., 2020, for summary).

We also compare our results specifically to Kraaijenbrink et al. (2017), which is the only regional study available so far that
explicitly accounted for the effect of debris cover. The study used remote sensing data to determine the spatial distribution of debris, as well as debris surface-temperature to estimate debris thickness and its relation to ice melt. Debris-cover extent and thickness was considered to be static, similarly to the case shown in Fig. 9 (grey line). Also, Kraaijenbrink et al. (2017) did not simulate ice flow explicitly, and did not calibrate the mass balance model against glacier-specific observations. Our results for HMA's total glacier volume loss during 2020-2100 are slightly more negative (between 5 and 11% for SSP126 and SSP585,
respectively) than the projections of GlacierMIP2. They also project more mass loss than Kraaijenbrink et al. (2017), with differences between 17 and 13% for SSP126 and SSP585, respectively. Finally, our results are between 4% less negative and 2% more negative for SSP126 and SSP585, respectively, than the mean result of Edwards et al. (2021) using the same climate forcing data (Fig. 11).


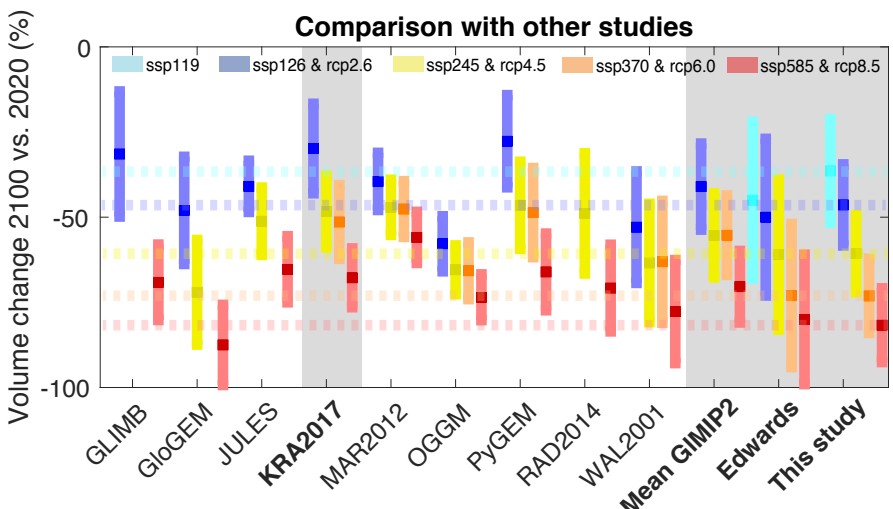

**Figure 11.** Comparison of modelled volume changes with values from Marzeion et al. (2020) and Edwards et al. (2021). Changes are expressed with respect to the 2020 baseline. From left to right, the abbreviations (GLIMB, GloGEM, JULES, KRA2017, MAR2012, OGGM, PyGEM, RAD2014, WAL2001) stand for Sakai and Fujita (2017); Huss and Hock (2015); Shannon et al. (2019); Kraaijenbrink et al. (2017); Marzeion et al. (2012); Maussion et al. (2019); Rounce et al. (2020); Radič and Hock (2014); Van de Wal and Wild (2001).

We suspect that the somewhat larger mass loss compared to GlacierMIP2 can be attributed to higher climate sensitivity of the CMIP6 GCMs used in our study compared to the CMIP5 GCMs (Wyser et al., 2020) used in Marzeion et al. (2020). In general, the explicit inclusion of debris-cover dynamics does not result in fundamentally different volume projections at the regional scale. Our results, however, permit to better represent the transient processes that control the changes at the scale of individual glaciers. For a detailed comparison between our results and various site-specific studies for HMA, we refer the reader to the Supplementary Material.

## 8   Conclusions

In this study, we presented a new module for simulating the spatio-temporal evolution of supraglacial debris. We implemented the new approach into the glacier model GloGEMflow, and showed its applicability from the single-glacier to the regional scale. By relying on glacier-specific Østrem curves – i.e. functions that characterize the relation between debris-cover thickness and ablation rates – the module accounts for the enhanced and reduced melting caused by debris thinner or thicker than 3-4 cm, respectively. The temporal evolution of both the spatial distribution and the thickness of the supraglacial debris is controlled by the glacier's mass balance, equilibrium line altitude, and pre-existing debris properties. The mass-balance module was calibrated through glacier-specific geodetic ice volume changes, while the debris-evolution module was calibrated and evaluated independently with remote sensing observations. The model was applied to all HMA glaciers with two modalities: one where the supraglacial debris cover is accounted for explicitly, and one where this is only done implicitly, i.e. by using the




same climatic conditions but by re-calibrating the parameters of the mass-balance module to match observed volume change
whilst pretending all glaciers to be debris-free.

When explicitly modelling debris, we found that both the debris-covered fraction and the local debris thickness will increase
in the future. This is related to the ongoing atmospheric warming, with larger debris-cover changes projected for scenarios of
higher warming. Averaged over the transient glacier area, the mean debris thickness will only slightly increase. This is due the
expansion of areas where new, thin debris is expected to form.

Perhaps surprisingly, explicitly modelling debris-cover evolution has only a small effect on the regional-scale glacier volume
and area evolution. The difference to the case in which the debris cover is modelled implicitly is below 3 %. On the one hand,
this is due to the fact that the majority of HMA's glaciers are debris-free. The regional volume and area evolution is an
average over a vast number of glaciers, with individual glacier-specific signals cancelling out each other due to site-specific
evolution of debris and geometry for each glacier. This does not mean that the glaciers' debris cover can be neglected, or that it
is encouraged to account for debris only implicitly. At the glacier-specific scale, in fact, the difference between explicitly and
implicitly modelling debris cover becomes important. We found, for example, that explicit modelling of debris can significantly
decrease the mass balance gradient of a given glacier. This results in turn in a reduced mass turnover, with consequences for
the future evolution of the glacier's geometry or the modelled surface ice velocities. At the level of individual glaciers, such
quantities are important, as they can have implications on e.g. water availability and natural hazards.

Based on the above appreciation, we encourage to explicitly account for the temporal evolution of supraglacial debris when
modelling debris-covered glaciers. We also suggest to do so at the glacier-specific scale, but also at the regional scale – espe-
cially when addressing regions that feature a significant share of debris-covered glaciers. We also promote further investigations
directed to the past evolution of debris-covered area and thickness. This would be important for acquiring further knowledge
about the processes controlling their evolution, as well as for better constraining some of the necessary model parameters. We
suggest that both remote-sensing observations, as well as field-based methods will be valuable in this respect.

*Code availability.* Code and data availability will be made available upon request

*Author contributions.* LC, DF, MH and HZ conceived the study. LC performed the numerical modelling with support from MH, ESM, MJM,
HZ, FP and DF. The original code was developed by MH (mass balance and debris evolution modules) and by HZ (ice flow module). MJM
and ESM produced the estimates of debris thickness and the glacier-specific Østrem curves. ESM produced the debris cover map from the
multiple Landsat satellite images. LC wrote the manuscript and produced the figures, with contributions from all other co-authors.

*Competing interests.* There are no competing interests.



*Acknowledgements.* We are grateful to the Swiss National Science Foundation, and for the funding of project Nr. 184634 in particular. HZ acknowledges the funding from a Marie Skłodowska-Curie Individual Fellowship (Grant 799904) and from the Fonds de la Recherche
Scientifique (FNRS postdoctoral fellowship). This publication was supported by PROTECT and RAVEN, which has received funding from the European Union's Horizon 2020 research and innovation programme under grant agreement Nso. 869304 and 772751, respectively. We thank the RGI consortium for the global glacier inventory data. We acknowledge the ECMWF for the ERA-5 re-analysis, and CMIP for the GCM outputs. We also thank the WGMS for providing mass balance and length change measurements. We thank also Amaury Dehecq, who provided the Hexagon images.



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
