# Peer review of "Modelling supraglacial debris-cover evolution from the single glacier to the regional scale: an application to High Mountain Asia"

_The Cryosphere, 2021_

## Referee Comment (RC2)

Review of Compagno et al, Modelling supraglacial debris-cover evolution from the single glacier to the regional scale: an application to High Mountain Asia

By Leif Anderson

**Overview**

This is an interesting study that simulates the expansion and thickening of debris and its affect on the response of debris-covered glaciers to climate change in HMA. There are new, simple parameterizations presented to represent the expansion and thickening of debris for application on both the individual and regional scales. It is shown that based on these parameterizations of debris thickness change and the *assumptions held within* that debris cover plays a minor role in glacier-wide mass balance evolution for simulations of HMA glaciers run out to 2100.

I applaud the immense effort put into this work and the novel contributions made. There are a number of exciting inferences and conclusions. But there are some major issues that need to be addressed before it is ready for publication.

**Major comments**

*Correct citing of Equation 1*
Some important citations are omitted. Some of which need to be cited or (I hate to have to say it) plagiarism is occurring. I am directly referring to Eq. (1) which was derived in its exact form by Anderson and Anderson (2016). This model is also presented in detail in the debris cover melt model intercomparison project manuscript (Pellicotti et al., in prep). This form of Østrem's curve is also referred to as the *hyper-fit model* by (Anderson et al., 2021a, b). I am personally glad it is useful it but *please* cite it appropriately.

*Neglected role of surface velocity in thickening and expanding debris*
The mass conservation equation for surface debris thickness change in time on a glacier surface includes debris melt out and dynamic re-distribution of debris:

$$\frac{\partial h_{debris}}{\partial t} = \frac{C\dot{b}}{(1-\phi)\rho_r} - \frac{\partial u h_{debris}}{\partial x} - \frac{\partial v h_{debris}}{\partial y} \qquad (3)$$

where $C$ is near-surface englacial debris concentration, $\varphi$ is the porosity of the debris, $\rho_r$ is the density of the rock composing the debris, and $u$ is surface velocity in the $x$-direction and $v$ is surface

Debris melt out is represented by the first term on the right and dynamic re-distribution of debris is represented by the second two terms on the right. This form is taken form Anderson et al. (2021b).

The parameterizations for debris thickness change presented here (Eqs. 4 and 6) do not include the effect of ice dynamics in changing debris extent or thickness. Equation 6 in this paper really only takes into account the debris melt out term in that debris thickness change is directly related to the melt rate as it varies in time by a factor (Eq. 4 is very similar):

$$h_{z,t} = h_{z,t-1} + abs(b_{z,t}) \cdot \overline{B_{(t-9,t)}} \cdot (-1) \cdot \overline{h_0} \cdot c_{thickening}, \quad \text{if } \gamma_{z,t-1} = 0 \tag{6}$$

where $h_{z,t}$ is the debris thickness for elevation $z$ and time $t$. $c_{thickening}$ is a calibration parameter for the debris-cover thickness evolution, constrained based on observations (see section 4.2). As for lateral debris expansion, the local mass balance $b_{z,t}$ relates linearly to debris thickness change. Higher melt rates will lead to faster debris thickening, thus implicitly assuming that debris concentrations within the ice are homogeneous. The long-term glacier-wide mass balance $\overline{B_{(t-9,t)}}$ mimics ice-dynamical processes. It leads to constant debris thickness for steady-state conditions ($\overline{B_{(t-9,t)}} = 0$), and to decreasing local debris thickness for consistent mass balances, thus mimicking the evacuation of debris with enhanced flow. This is in line with the few direct observations that are available (e.g. Gibson et al., 2017; Verhaegen et al., 2020). $\overline{h_0}$ is the mean debris thickness of the glacier at the inventory year. It parametrizes the effect that glaciers with a low mean debris thickness will thicken slower compared to glaciers with a high mean debris thickness. This is motivated by the assumption that glaciers with thick debris are likely to have a higher englacial debris concentration, indicative for high debris supplies from the surroundings.

Anderson and Anderson (2018) notes, using a theoretical analysis, that debris thickness patterns are strongly controlled by the inevitable decline of surface velocity down glacier. This is also basically outlined by Kirkbride (2000) and further supported by the modeling presented in Ferguson and Vieli (2021). I have not seen a compelling a reason in this manuscript why surface velocities should be neglected when considering the evolution of debris covers.

Furthermore, Anderson et al. (2021b) presents evidence that the dynamic effects of debris thickening (debris advection and debris compression) are unavoidably important for the thickening of debris in response to climate change. The dynamic affects on debris thickness are especially important where surface velocities are low and debris already tends to be thick. Meaning that debris might thicken substantially right where the model/parameterizations presented here is not accounting for it. This is because the thickening parameter is tuned with debris thickness estimates from the upglacier end of debris covers.

Anderson et al. (2021b) also shows how changes in flow patterns can change debris extent (I do believe this is mentioned breifly). I recognize that Anderson et al. (2021b) was published as this paper was coming out in TCD, but the work has direct implications for the parameterizations presented in this study.

I don't think the neglect of the role of ice dynamics makes this work invalid, rather the fact that the dynamical terms in the debris conservation equation are neglected should really *be discussed* and *stated very clearly in this paper*. Right now half of the continuity equation for debris change is assumed to be negligible in this manuscript, but the affect of surface velocities on debris thickness is not negligible even where surface velocities are low.

It is difficult to evaluate this work without reading McCarthy et al., submitted and understanding how the debris thicknesses across HMA were estimated. I am not sure where to access that manuscript. How many validation data points are used to evaluate the debris thickness estimates in that study? What do the debris thickness patterns look like? In some way this pre-requisite work needs to be made available be thorough description here or elsewhere.

**Minor comments**

There are places in the manuscript where the modelling results seem to be presented as reality but are really still just modeling results that rely on all of the assumptions inherent to the model design. More care should be taken to avoid overstatements.

I also wonder: How do the debris change evaluations change if the evaluating datasets are not used in the tuning process?

**Line-by-line comments**

Line 3: You could remove 'potential' here

Line 5-8: The sentence should be split in two as it is hard to follow as written.

Line 10: 'previous projections' would maybe be better here.

Line 15: no need for a '-' between debris and cover.

Line 48-51 see the inversion for debris thickness change by Anderson et al. (2021b) as an example of debris thickening. This paper also highlights how the change in direction in flow can lead to debris expansion.

Line 86. remove plural from 'glaciers'

Line 97. What do the variables represent beyond free parameters? I am surprised that Anderson and Anderson (2016) are not cited here. As that work originally derived this form of Østrem curve and discussed what these free parameters represent in detail. The model is called 'Hyper-fit' in Anderson et al., (2021a) and Anderson et al. (2021b). I hate to have to say this by as the text is written this is plagiarism. Please cite this appropriately.

Figure 3. The text in the figure is not legible in places.

Line 205. not sure what 'from the surroundings' means here.

Line 207. Deline (2005) is a valuable citation here.

**Section 3.2.2** this is an interesting parameterization but it should be stated that you *assume* that debris expansion is directly related to ELA change. As far as I am aware there are no datasets that show this as a direct relationship. Snowline change on glaciers is instantaneous but debris melt out does not need to be. This relationship is dependent on where englacial debris is present within the glacier.

Line 221. "As for the lateral expansion of debris, the evolution of debris thickness is linked to internal debris concentration and glacier mass balance (e.g. Gibson et al., 2017; Mölg et al., 2019; Verhaegen et al., 2020)."

Please see Anderson et al. (2021b) for a detailed process-based simulation of debris thickening that shows the importance of debris advection and compression (both highly dependent on surface velocities) as well as debris melt out.

Line 221. Also the change in surface velocity of the glacier: see Anderson et al (2021b)

252. 'w.e.a −1' add a space.

270. At this point I had forgotten what S1 was. Might be helpful for readers to remind them here?

**Section 4.2.2** This is a clever approach but again this is emphasizing the role of debris melt out as the only process that causes debris thickening. It is again hard to know though the validity of the McCarthy dataset without access to the errors from in situ debris thickness measurements.

**Section 5.2** It would be helpful to remind the reader where the pre-2000s climate forcings are coming from for the evaluation of the debris change. Maybe just re-state it or reference the section.

313. So you evaluate the lateral expansion parameretrization against the data that you used to tune it? I wonder how the parameterization works on glaciers that are independent of the tuning dataset?

344-346. This is an overstatement. Being off by 10 cm or ~20 cm of debris thickness, using the h_star values (same as (k_debris) in Equation 1) (the debris thickness change needed to reduce sub-debris melt rates by 50%) from A and A (2106) mean sub-debris melt rates are off by 50 to 100% or 200 to 400%. This uses h_star values = 5 cm and 10 cm. These percentages will be even bigger with melt amplification effects included. Since the evaluation of the debris thickness change estimates are coming from the upglacier end of debris covers the errors are actually quite large.

Figure 7. It would be helpful for the reader if the x axis was extended beyond +-200 m

358. In the ablation zone or across the whole glacier?

439-440. "In fact, accounting for the debris cover explicitly enables the model to correctly represent the driving processes, rather than compensating the lack of model capabilities through a suitable parameter choice."

I suggest that this be re-written as it is a significant overstatement from my reading. The explicit model presented here neglects the role of debris advection and compression and is evaluated with highly uncertain modelled debris thickness estimates. I would replace 'correctly' throughout this paragraph.

446. Citations would be helpful here.

Section 7.2 Nice to have this clear statement of the sensitivity!

457. Also from the assumptions held within each parameterization.

Section 7.3 Interesting analysis/results.

519. when only debris melt out is included.

528. typo

References

Anderson, L. S. and Anderson, R. S.: Modeling debris-covered glaciers: response to steady debris deposition, 10, 1105–1124, https://doi.org/10.5194/tc-10-1105-2016, 2016.

Anderson, L. S., Armstrong, W. H., Anderson, R. S., and Buri, P.: Debris cover and the thinning of Kennicott Glacier, Alaska: in situ measurements, automated ice cliff delineation and distributed melt estimates, The Cryosphere, 15, 265–282, https://doi.org/10.5194/tc-15-265-2021, 2021a.

Anderson, L. S., Armstrong, W. H., Anderson, R. S., Scherler, D., and Petersen, E.: The Causes of Debris-Covered Glacier Thinning: Evidence for the Importance of Ice Dynamics From Kennicott Glacier, Alaska, 9, 19, 2021b.

Deline, P.: Change in surface debris cover on Mont Blanc massif glaciers after the'Little Ice Age' termination, 15, 302–309, https://doi.org/10.1191/0959683605hl809rr, 2005.

Kirkbride, M. P.: Ice-marginal geomorphology and Holocene expansion of debris-covered Tasman Glacier, New Zealand, in: Debris-Covered Glaciers, Proceedings of a workshop at Seattle, Washington, USA September 2000, 211–217, 2000.

---

## Author Comment (AC2)

**Author's response to the comments received for tc-2021-31**

The following pages contain a point-by-point reply to the comments provided by the two referees that reviewed our first submission (TC-2021-31)

Each of the referee's comment (**RC**) is numbered. If a comment contained several points, we numbered them, and address them individually in our author replies (**AR**).

**REVIEWER 2 - LEIF S. ANDERSON**

**Overview**

[RC 2.01] (i) This is an interesting study that simulates the expansion and thickening of debris and its affect on the response of debris-covered glaciers to climate change in HMA. There are new, simple parameterizations presented to represent the expansion and thickening of debris for application on both the individual and regional scales. It is shown that based on these parameterizations of debris thickness change and the assumptions held within that debris cover plays a minor role in glacier-wide mass balance evolution for simulations of HMA glaciers run out to 2100.

I applaud the immense effort put into this work and the novel contributions made. There are a number of exciting inferences and conclusions. (ii) But there are some major issues that need to be addressed before it is ready for publication.

[AC 2.01] (i) We thank the reviewer for the positive feedback and for the thorough review. (ii) Below, we have addressed all issued raised by the reviewer. The manuscript was updated accordingly.

**Major comments**

[RC 2.02] *Correct citing of Equation 1*
Some important citations are omitted. Some of which need to be cited or (I hate to have to say it) plagiarism is occurring. I am directly referring to Eq. (1) which was derived in its exact form by Anderson and Anderson (2016). This model is also presented in detail in the debris cover melt model intercomparison project manuscript (Pellicotti et al., in prep). This form of Ostrem's curve is also referred to as the hyper-fit model by (Anderson et al., 2021a, b). I am personally glad it is useful it but please cite it appropriately.

[AC 2.02] It was far from our intentions to commit plagiarism, and we are very sorry that the references pointed out were missing. We now make mention of the reviewer's work when introducing Eq. 1:

l. 106-108: '*Note that equation 1 has similarities with the Hyper-fit model of Anderson and Anderson (2016), and Anderson et al. (2021a, b), although we note that the two approaches differ in the number of parameters and their interpretation.'*

[RC 2.03] *Neglected role of surface velocity in thickening and expanding debris*
(i) The mass conservation equation for surface debris thickness change in time on a glacier surface includes debris melt out and dynamic re-distribution of debris:

$$\frac{\partial h_{debris}}{\partial t} = \frac{C\dot{b}}{(1-\phi)\rho_r} - \frac{\partial u h_{debris}}{\partial x} - \frac{\partial v h_{debris}}{\partial y} \qquad (3)$$

where $C$ is near-surface englacial debris concentration, $\varphi$ is the porosity of the debris, $\rho_r$ is the density of the rock composing the debris, and $u$ is surface velocity in the $x$-direction and $v$ is surface

Debris melt out is represented by the first term on the right and dynamic re-distribution of debris is represented by the second two terms on the right. This form is taken form Anderson et al. (2021b).

The parameterizations for debris thickness change presented here (Eqs. 4 and 6) do not include the effect of ice dynamics in changing debris extent or thickness. Equation 6 in this paper really only takes into account the debris melt out term in that debris thickness change is directly related to the melt rate as it varies in time by a factor (Eq. 4 is very similar):

$$h_{z,t} = h_{z,t-1} + abs(b_{z,t}) \cdot \overline{B_{(t-9,t)}} \cdot (-1) \cdot \overline{h_0} \cdot c_{thickening}, \quad \text{if } \gamma_{z,t-1} = 0 \qquad (6)$$

where $h_{z,t}$ is the debris thickness for elevation $z$ and time $t$. $c_{thickening}$ is a calibration parameter for the debris-cover thickness evolution, constrained based on observations (see section 4.2). As for lateral debris expansion, the local mass balance $b_{z,t}$ relates linearly to debris thickness change. Higher melt rates will lead to faster debris thickening, thus implicitly assuming that debris concentrations within the ice are homogeneous. The long-term glacier-wide mass balance $\overline{B_{(t-9,t)}}$ mimics ice-dynamical processes. It leads to constant debris thickness for steady-state conditions ($\overline{B_{(t-9,t)}} = 0$), and to decreasing local debris thickness for consistent mass balances, thus mimicking the evacuation of debris with enhanced flow. This is in line with the few direct observations that are available (e.g. Gibson et al., 2017; Verhaegen et al., 2020). $\overline{h_0}$ is the mean debris thickness of the glacier at the inventory year. It parametrizes the effect that glaciers with a low mean debris thickness will thicken slower compared to glaciers with a high mean debris thickness. This is motivated by the assumption that glaciers with thick debris are likely to have a higher englacial debris concentration, indicative for high debris supplies from the surroundings.

**(ii)** Anderson and Anderson (2018) notes, using a theoretical analysis, that debris thickness patterns are strongly controlled by the inevitable decline of surface velocity down glacier. This is also basically outlined by Kirkbride (2000) and further supported by the modeling presented in Ferguson and Vieli (2021). I have not seen a compelling a reason in this manuscript why surface velocities should be neglected when considering the evolution of debris covers.

**(iii)** Furthermore, Anderson et al. (2021b) presents evidence that the dynamic effects of debris thickening (debris advection and debris compression) are unavoidably important for the thickening of debris in response to climate change. The dynamic affects on debris thickness are especially important where surface velocities are low and debris already tends to be thick. Meaning that debris might thicken substantially right where the model/parameterizations presented here is not accounting for it. This is because the thickening parameter is tuned with debris thickness estimates from the upglacier end of debris covers.

**(iv)** Anderson et al. (2021b) also shows how changes in flow patterns can change debris extent (I do believe this is mentioned breifly). I recognize that Anderson et al. (2021b) was published as this paper was coming out in TCD, but the work has direct implications for the parameterizations presented in this study.

**(v)** I don't think the neglect of the role of ice dynamics makes this work invalid, rather the fact that the dynamical terms in the debris conservation equation are neglected should really be discussed and stated very clearly in this paper. Right now half of the continuity equation for debris change is assumed to be negligible in this manuscript, but the affect of surface velocities on debris thickness is not negligible even where surface velocities are low.

**(vi)** It is difficult to evaluate this work without reading McCarthy et al., submitted and understanding how the debris thicknesses across HMA were estimated. I am not sure where to access that manuscript. How many validation data points are used to evaluate the debris thickness estimates in that study? What do the debris thickness patterns look like? In some way this pre-requisite work needs to be made available be thorough description here or elsewhere.

**[AC 2.03]**

(i)  We thank the reviewer for providing us with the formula. From a theoretical point of view, it makes certainly sense to divide the debris thickness evolution into debris melt out and dynamic re-distribution of debris. However, note that this formula also assumes that the englacial debris concentration, the porosity of the debris, and the density of the rock composing the debris are completely known. At the regional scale (and mostly even at the local scale), such data are not available, leaving the formula with a number of unknowns. In our approach (Eq. 4 and Eq. 7), we do not account explicitly for the effect of surface velocity change. This is because during calibration it would be impossible to divide the observed debris-thickness change into the two necessary components (i.e. debris melt out and dynamic redistribution of debris). The latter would only be possible if the englacial debris concentration, the porosity of debris and the density of the rock composing the debris were known for each of the modelled glaciers. We emphasize that the physics-based analyses of debris evolution are vital for advancing understanding, but this is a reasonable first approach to represent the major changes in debris thickness patterns that should be expected, consistent with the modelling scope (climate projections).

That said, the dynamical redistribution of debris is not entirely omitted with our parameterizations. Rather, it is implicitly accounted for thanks to our calibration strategy. Indeed, the calibration of $c_{thickening}$ (and also $c_{lateral}$) with observations ensures that changes due to both debris melt out as well as dynamic re-distribution of debris are reproduced.

(ii)  See above: our approach accounts for dynamical re-distribution implicitly (cf. 2.03 i). We agree that taking this into account explicitly (as done in Kirkbride (2000), Andrerson and Anderson (2018), and Ferguson and Vieli (2021), for example) would be ideal. It is impossible however at the scale of our investigation. Indeed that the mentioned studies either refer to synthetic glaciers (Andrerson and Anderson (2018), Ferguson and Vieli (2021)), or were only applied to extremely well-studied individual glaciers (Tasman Glacier in the case of Kirkbride, 2000). Applying similar methods to more than 6'000 glaciers in combination with climate forcing taken from more than 50 global circulation models would be computationally impossible, and severely limited by the lack of necessary input data. In brief, we argue that some simplifications are indispensable for regional-scale modelling, and this is exactly what our approach does.

(iii)  This is not true. With our parameterization, the debris thickness will growth at a faster rate for areas with already thick debris cover compared to where debris is thinner (the debris thickness of a given elevation band is accounted for in in the second term of Equation 7). The same applies for Equation 4. Again, we note that our parameterization takes the effect of dynamic redistribution of debris into account implicitly. For an example of the results, see Fig. 8a, S6a and S6(b).

(iv)  We added a reference to the mentioned paper, and we reformulated l. 262-265 to make our parameterization clearer:
*'Combined with $b_{(z,t)}$, the long-term glacier-wide mass balance $B_{(t–9,t)}$ mimics ice-dynamical processes. It leads to constant debris thickness for steady-state conditions ($B_{(t–9,t)}$=0), to a*

*growth of debris thickness with negative mass balances (thus mimicking dynamic re-distribution of debris and its compression, e.g. Kirkbride, 2000; Anderson et al., 2021b; Ferguson and Vieli, 2021), and to decreasing debris thickness for positive mass balances (thus mimicking the evacuation of debris with enhanced flow).'*

(v) We added a paragraph in the discussion section to discuss this topic:
l.497-503:*'Additional uncertainties arise also from the parameterizations themselves (Eqs. 4, 5 and 7). In a simplified but realistic way, our approach parameterizes the evolution of debris cover on glaciers, and it is based on debris-evolution patterns observed in the last decades. We acknowledge that this is not the only way that debris evolution could be parameterized. Accounting for debris re-distribution dynamics explicitly (as opposed to implicitly, see section 3.2.3) could be an option, as done by Anderson et al. (2021b), for example. We decided to include such processes implicitly (1) because the absence of data to calibrate and validate a more complex parameterization at regional/global scales, and (2) due to the small sensitivity of volume and area evolution to changes in the debris-cover evolution when considering the entire region (see previous paragraph).'*

(vi) We admit that it was our mistake not to provide the reviewers with the manuscript in review by McCarthy et al. The manuscript is now available at https://doi.org/10.31223/X5WW5B . Note that we reformulated the section referring to McCarthy et al. (in review) to clarify the methods (cf. AR 1.01). We used 148007 individual data points of 13 glaciers. However, the majority of these points come from closely spaced GPR measurements made on Ngozumpa Glacier. The median number of in-situ data points per glacier is 37. For the thickness pattern, there are typically an increase in debris thickness down-glacier toward the terminus, but there is also substantial variability between glaciers with different morphologies.

**Minor comments**

**[RC 2.04] (a)** There are places in the manuscript where the modelling results seem to be presented as reality but are really still just modeling results that rely on all of the assumptions inherent to the model design. More care should be taken to avoid overstatements.
**(b)** I also wonder: How do the debris change evaluations change if the evaluating datasets are not used in the tuning process?

**[AR 2.04]**
**(a)** We reworded many sentences in the results section, in order to clarify that we are presenting modelling results based on parameterizations, and not observations.

E.g.:

l 379-380: *'Figure 8a shows a profile view of the glacier and debris-cover evolution of Langtang Glacier according to our model results and SSP245.'*

l 394-395: *'As a result of the projected up-glacier migration, instead, the maximum elevation with supraglacial debris would increase by 280 m for the same time period.'*

l 397-398: *'With higher mass loss, our model results show that both debris-covered area and debris thickness increase, thus reducing ice melt.'*

l 409: *'Compared to 2020, the modelled mean debris thickness of Langtang Glacier is expected to […]'*

l 437-438: *'Locally, however, much higher debris thickness increases are modelled but these are offset at many places by the overall reduction in glacier area.'*

l 469-475: reformulated, see line by line comment AR 2.26.

**(b)** For calibration and validation different datasets are used. We imagine that here the reviewer is referring to the calibration and evaluation of the lateral spread out of debris cover (see **RC 2.22**).

In this case, it is correct that for the evaluation we use a subset of glaciers also used for the calibration. However, the analysis is initialized with different datasets and time periods. In the calibration, we used three to five Landsat scenes between 1989 and 2020. The model is initialized with debris extents of each of these scenes, and then compared with the older scenes. In the evaluation, we use Hexagon images of around 1974 as initial condition, and then we compare the model results with the Landsat scenes.

To investigate the sensitivity to these choices, we re-calibrated the model omitting all glaciers used in the evaluation. By doing so, the mean of the parameter $c_{lateral}$ is again found to be c_lateral=2.0 (as already used before in the manuscript). This shows that the results would not change if the glaciers used in the evaluation would not be used in the calibration, see figure below).

[Figure]

Figure AR1: Distribution of the extension tuning factor $c_{lateral}$ resulting in the lowest misfit between observed and modelled debris-fraction evolution (given per number of glaciers). In contrast to Figure 4, glaciers used in the evaluation are omitted during calibration. The grey dashed line shows the mean value while the grey rectangle shows values within the 0.25 and 0.75 quartiles.

**Line-by-line comments**

**[RC 2.05]** Line 3: You could remove 'potential' here

**[AR 2.05]** Removed

**[RC 2.06]** Line 5-8: The sentence should be split in two as it is hard to follow as written.

**[AR 2.06]** We split the sentence:
L. 5-8: *'The module is initialized with both glacier-specific observations of the debris' spatial distribution and estimates of debris thickness. These data sets account for the fact that debris can either enhance or reduce surface melt depending on thickness. Our model approach also enables representing the spatio-temporal evolution of debris extent and thickness.'*

**[RC 2.07]** Line 10: 'previous projections' would maybe be better here.

**[AR 2.07]** We rewrote the sentence into:
*'Explicitly accounting for debris cover has only a minor effect on the projected mass loss, which is in line with previous projections. Despite this small effect, we argue that the improved process representation is of added value when aiming at capturing intra-glacier scales, i.e. spatial mass balance distribution.'*

**[RC 2.08]** Line 15: no need for a '-' between debris and cover.

**[AR 2.08]** Removed

**[RC 2.09]** Line 48-51 see the inversion for debris thickness change by Anderson et al. (2021b) as an example of debris thickening. This paper also highlights how the change in direction in flow can lead to debris expansion.

**[AR 2.09]** cf. AR 2.03

**[RC 2.10]** Line 86. remove plural from 'glaciers'

**[AR 2.10]** Corrected

**[RC 2.11]** Line 97. What do the variables represent beyond free parameters? I am surprised that Anderson and Anderson (2016) are not cited here. As that work originally derived this form of Ostrem curve and discussed what these free parameters represent in detail. The model is called 'Hyper-fit' in Anderson et al., (2021a) and Anderson et al. (2021b). I hate to have to say this by as the text is written this is plagiarism. Please cite this appropriately.

**[AR 2.11]** The two parameters have been fitted to results of an energy-balance model. For more details see AR 2.02. We have now acknowledged that this is similar to the 'Hyper-fit' model, although we derived, formulated, and interpreted the form of the equation independently.

**[RC 2.12]** Figure 3. The text in the figure is not legible in places.

**[AR 2.12]** We increase the font size and added a black border to the text in order to make it more legible.

[Figure]

**[RC 2.13]** Line 205. not sure what 'from the surroundings' means here.

**[AR 2.13]** We re-wrote the sentence into:
l. 223-224: *'Areas with abundant debris cover may grow faster due to enhanced debris supply from melt-out, or due to ice flow changes (Anderson, 2000; Anderson et al. 2021b)'*

**[RC 2.14]** Line 207. Deline (2005) is a valuable citation here.

**[AR 2.14]** We added the suggested reference in the manuscript.

**[RC 2.15] (i)** Section 3.2.2 this is an interesting parameterization but it should be stated that you assume that debris expansion is directly related to ELA change. **(ii)** As far as I am aware there are no datasets that show this as a direct relationship. Snowline change on glaciers is instantaneous but debris melt out does not need to be. **(iii)** This relationship is dependent on where englacial debris is present within the glacier.

**[AR 2.15] (i)** Yes, with this parametrization we assume that the debris expansion is directly related to the ELA change. We added this information to the manuscript:
l. 228-229: *'We assume that this is related to the rise of the ELA and to the melt-out of debris in areas that transit from the accumulation to the ablation zone (Anderson, 2000).'*

**(ii)** We agree that snowline change on glaciers is instantaneous but debris melt out does not need to be, therefore we compute the moving average of the ELA change over the last 10 years.

**(iii)** The parameterization expands the debris upglacier only from where it already exists (i.e. not in the middle of a clean ice glacier), thus accounting for the fact that englacial debris must not be present in every glacier or everywhere on the glacier.
Note that we reformulated also other parts of the section, as requested by reviewer 1 (cf. AR 1.16)

**[RC 2.16]** Line 221. "As for the lateral expansion of debris, the evolution of debris thickness is linked to internal debris concentration and glacier mass balance (e.g. Gibson et al., 2017; Mölg et al., 2019; Verhaegen et al., 2020)."
Please see Anderson et al. (2021b) for a detailed process-based simulation of debris thickening that shows the importance of debris advection and compression (both highly dependent on surface velocities) as well as debris melt out.

**[AR 2.16]** We added this information. We also discussed it in detail in AR 2.04.
l. 242-244: *'As for the lateral expansion of debris, the evolution is linked to internal debris concentration and glacier mass balance (e.g. Gibson et al., 2017; Mölg et al., 2019; Verhaegen et al., 2020), as well as to changes in ice flow velocity (e.g. Anderson et al., 2021b)'*

**[RC 2.17]** Line 221. Also the change in surface velocity of the glacier: see Anderson et al (2021b)

**[AR 2.17]** cf. AR 2.03 and cf. AR 2.16

**[RC 2.18]** 252. 'w.e.a −1' add a space.

**[AR 2.18]** We added the spaces (w.e. a$^{-1}$)

**[RC 2.19]** 270. At this point I had forgotten what S1 was. Might be helpful for readers to remind them here?

**[AR 2.19]** We added what set S1 is:
'*To determine $c_{lateral}$, we use the debris-cover observations obtained from the Landsat scenes (set S1, composed of 55 glaciers with debris. See section 2 and e.g. Fig. 4a).*'

**[RC 2.20]** Section 4.2.2 This is a clever approach but again this is emphasizing the role of debris melt out as the only process that causes debris thickening. It is again hard to know though the validity of the McCarthy dataset without access to the errors from in situ debris thickness measurements.

**[AR 2.20]** Please refer to AR 2.03.

**[RC 2.21]** Section 5.2 It would be helpful to remind the reader where the pre-2000s climate forcings are coming from for the evaluation of the debris change. Maybe just re-state it or reference the section.

**[AR 2.21]** Before 2020 we always use ERA5 reanalysis. We added this information in the manuscript.
l. 339-341: '*To evaluate the parametrization for lateral debris expansion, 18 glaciers (set S1 with Hexagon satellite observations) within three sub-regions (Central Himalaya, West Himalaya and West Tien Shan) are simulated from 1974 to 2020 (forcing the model with ERA5 climate).*'

**[RC 2.22]** 313. So you evaluate the lateral expansion parameretrization against the data that you used to tune it? I wonder how the parameterization works on glaciers that are independent of the tuning dataset?

**[AR 2.22]** Please see AR 2.04 b

**[RC 2.23]** 344-346. This is an overstatement. Being off by 10 cm or ~20 cm of debris thickness, using the h_star values (same as (k_debris) in Equation 1) (the debris thickness change needed to reduce sub-debris melt rates by 50%) from A and A (2106) mean sub-debris melt rates are off by 50 to 100% or 200 to 400%. This uses h_star values = 5 cm and 10 cm. These percentages will be even bigger with melt amplification effects included. Since the evaluation of the debris thickness change estimates are coming
from the upglacier end of debris covers the errors are actually quite large.

**[AR 2.23]** We need evaluation data for debris thickness changes and englacial concentration that are representative for different parts of debris-covered glaciers, including the debris emergence region (Stewart et al., 2020) as well as the terminus. It is noteworthy that a single debris concentration is a key step forward for this (e.g. Anderson et al, 2021b) but insufficient given the variations in englacial debris concentration (e.g. Anderson and Anderson, 2018; Miles et al, 2021).

**[RC 2.24]** Figure 7. It would be helpful for the reader if the x axis was extended beyond +-200 m

**[AR 2.24]** We changed the limit of the axis to +-250 m, highlighting that there is no data beyond this range.

[Figure]

[Figure]

**[RC 2.25]** 358. In the ablation zone or across the whole glacier?

**[AR 2.25]** In the ablation zone. We changed the sentence into:
L. 385-386: *'This leads to a nearly homogeneous downwasting of the ablation zone (e.g. Pellicciotti et al., 2015; Ragettli et al., 2016) rather than to a retreat of the terminus (e.g. Benn et al., 2012).'*

**[RC 2.26]** 439-440. "In fact, accounting for the debris cover explicitly enables the model to correctly represent the driving processes, rather than compensating the lack of model capabilities through a suitable parameter choice."
I suggest that this be re-written as it is a significant overstatement from my reading. The explicit model presented here neglects the role of debris advection and compression and is evaluated with highly uncertain modelled debris thickness estimates. I would replace 'correctly' throughout this paragraph.

**[AR 2.26]** We adapted the sentence in order to avoid the impression of an overstatement. As suggested by the reviewer, we also adapted the rest of this paragraph:
l.469-475: *'In fact, accounting for the debris cover explicitly enables the model to mimic the driving processes, rather than compensating the lack of model capabilities through a suitable parameter choice. This is important, especially when results other than area and volume changes are of interest. Indeed, quantities such as the local mass balance, the glaciers' ice flow velocity and mass turnover, the glacier's length change or water runoff, are only captured adequately when explicitly accounting for supraglacial debris and its temporal evolution. These quantities, in turn, have to be*

*modelled appropriately when aiming at anticipating other glacier-related processes, such as hazards from ice-dammed or proglacial lakes, or potential slope instabilities.'*

**[RC 2.27]** 446. Citations would be helpful here.

**[AR 2.27]** We added two citations: Kraaijenbrink et al. and 2017 and Rounce et al., 2021. l.450-476-479: *'Compared to the static representation of supraglacial debris cover that is presently included in some regional to global glacier models (e.g. Kraaijenbrink et al., 2017; Rounce et al., 2021), the expected increase in both debris-cover fraction and local debris thickness will enhance the insulating effects of the debris cover.*

**[RC 2.28]** Section 7.2 Nice to have this clear statement of the sensitivity!

**[AR 2.28]** We thank the reviewer.

**[RC 2.29]** 457. Also from the assumptions held within each parameterization.

**[AR 2.29]** True. We added a paragraph describing this (cf. AR 2.03 v)

**[RC 2.30]** Section 7.3 Interesting analysis/results.

**[AR 2.30]** We thank the reviewer.

**[RC 2.31]** 519. when only debris melt out is included.

**[AR 2.31]** We added 'with our approach' to clarify:
l. 557-558:*'Averaged over the transient glacier area, our approach anticipates the mean debris thickness to increase only slightly'*

**[RC 2.32]** 528. typo

**[AR 2.32]** We are somewhat embarrassed but can't find any typo at this line. Possibly it is a different line than former L. 528?

**REFERENCE:**
Compagno, L., Zekollari, H., Huss, M. & Farinotti, D. Limited impact of climate forcing products on future glacier evolution in Scandinavia and Iceland. *J. Glaciol.* **67**, 727–743 (2021b).

Stewart, L. G., Lavers, J. L., Grant, M. L., Puskic, P. S. & Bond, A. L. Seasonal ingestion of anthropogenic debris in an urban population of gulls. *Marine Pollution Bulletin* **160**, 111549 (2020).

---

## Author Response (AR1)

**Author's response to the comments received for tc-2021-31**

The following pages contain a point-by-point reply to the comments provided by the two referees that reviewed our first submission and to by the editor that commented our second submission.

Each of the referee's comment (**RC**) and editor's comment (**EC**) are numbered. If a comment contained several points, we numbered them, and address them individually in our author replies (**AR**).

**EDITOR - ANDREAS VIELI**

[EC 0.01] Comments to the author:

Dear Loris Compagno and co-authors,

Your manuscript received two very detailed and critical reviews but that both highlighted the novelty and innovation in the treatment of debris in regional/global scale glacier models and the related extensive modelling results/investigations and they confirm that the manuscript would be a very valuable contribution to TC. Although both referees were in general very positive, they also raised, besides the minor technical corrections, also a few major issues that should be improved before publication. In brief the most important points encompass

a) clarifications and more details in the methods (in particular the debris thickness data and the parametrized oestroem curve

b) issue of omitting dynamic debris thickening/thinning (velocity) effects (add at least a discussion on this)

c) better evaluate the role of debris cover inclusion (omitting, explicit, implicit)

d) more accurate reference to literature

Given the response of the authors, it seems that the authors seem to be able to address the points (or already have done so) raised by the referees well and that the revised manuscript has a good chance to be accepted. I therefore ask the authors to undertake the planned revisions and submit the revised document.

**[AR 0.01]** We thank the editor for the positive feedback. We will send the revised manuscript as instructed. Here below we have addressed all issues raised by the editor. The manuscript was updated accordingly.

**[EC 0.02]** A few more notes from me as the editor to your suggested revisions to the above more major points:

to a)

With regard to the requested more detailed explanation of the used oestroem curve parametrization by referee 1, there are still a few points unclear or missing in your suggested revisions that should maybe be already considered:

(i) - when explaining how the debris thickness maps of McCarthy were obtained it would be useful to mention the basic principle behind (e.g. inversion of mass conservation equation from remote sensing data (velocity fields, dh/dt), rather than just refer to the literature.

(ii) - clarify that parameters i\_debris and k\_debris are parameters that are determined FOR EACH GLACIER (I would actually also wonder what values you would get there, range, mean, median?).
(iii) - it would also be more explicit what the physical meaning of these parameters (in particular k\_debris) are (in words). I believe k\_debris is similar to the d\_0 value of Anderson 2016 (some sort of reference debris thickness), whereas as i\_debris is a pure calibration factor.

(iv) - the new Fig. S1 is very useful, but add in figure (or caption) what the parameter values are to it (I believe all green curves use the same h\_eff and h\_crit but a different k\_debris (how variable?).
(v) - the supposedly newly added 'unceratinty' in debris thickness in fig. 1 b)-d) is not really explained in the caption, I just see a +? and a -? Number after the debris thickness value, what do they refer to? The range? The uncertainty in both directions? Calrify at least in caption.

**[AR 0.02]**

(i) We added an explanation for the basic principle by which the debris thickness maps were obtained:

II. 98-100: 'In a nutshell, the inversion procedure iteratively solves for the debris thickness by using an energy balance model. The procedure uses DEMs, glacier ice thickness, surface velocity, debris proprieties, and meteorological forcing data as input, and uses them to calculate ice flux divergence and ice thinning rates. The debris thickness is then adjusted until modelled and observed ice-melt rates agree within a prescribed tolerance.'

**(ii) We added the information asked by editor:**

II. 110-112 ' [...] where b is the local surface mass balance, and  $i_{debris}$  and  $k_{debris}$  are glacierspecific calibration parameters without specific physical meaning. The the mean and 95% confidence interval for  $i_{debris}$  are -1.86 and [-7.62, -0.09], respectively. For  $k_{debris}$  the equivalent values are 0.10, and [0.01, 0.22].'

(iii) The parameters idebris and kdebris do not have a physical meaning. They are calibration factors used to fit the results of the energy balance model. We added this information in the manuscript (cf. AR 1.02 ii).

(iv) The editor is right,  $h_{eff}$  and  $h_{crit}$  are always the same whilst  $k_{debris}$  is glacier-specific (cf. AR1.02 ii). We added the value of  $k_{debris}$  in the Figure 3 as request by the editor.

'Figure 3: Schematic of the melt enhancement factor g (dimensionless) as a function of debris thickness for three different glaciers (green lines; "k" is the value of k\_debris calibrated for each glacier). The colored, dashed boxes show regions in which the different cases of Eq. 2 and 3 apply.'

(v) The values shown in Figure 1 (b-d) define the confidence interval estimated for the mean debris thickness of each glacier. We updated the caption as follows:

Fig1: '[...] hdebris is the mean debris-cover thickness with superscript and subscript values indicating its estimated confidence interval (note that the latter is not symmetric; cf. Section 2.2) [...]'

**[EC 0.03] to b)**

with regard to the neglectance of dynamic contribution to debris thickening/thinning (Referee 2) some discussion on additional uncertainties has been added. But this could be a bit more explicit in saying what the issue is (neglecting dynamic effects of debris thickening/thinning) and why it may be justified (indirectly included, etc... see you response to the review) to do it your way. It could just be a bit more on the point.

**[AR 0.03]** We slightly revised the manuscript, clarifying both what the issue is (neglecting dynamic effects of debris thickening/thinning) and that this effect is only implicitly accounted for in our approach. Similarly, we now explain why we only accounted for it implicitly, instead of explicitly. In particular, we modified the following sentences:

1 257-258: 'Combined with  $b_{(z,t)}$ , the long-term glacier-wide mass balance  $B_{(t-9,t)}$  implicitly mimics ice-dynamical processes (see section 7.2)'

1 504-509: 'Explicitly accounting for the dynamics of debris re-distribution (as opposed to implicitly, see Section 3.2.3) could be an alternative option. Indeed, where the ice flow velocity decreases, the local debris thickness increases and vice versa, due to the convergence of debris particles. Such an approach was followed by Anderson et al. (2021b), for example. We decided not to include such effects explicitly because (1) the absence of data to calibrate and validate a more complex parameterization at regional scales, and (2) the small sensitivity of regional-scale glacier volume and area to changes in debris cover (see Section 7.1).'

**REVIEWER 1 - BEN MARZEION**

**[RC 1.00] (i)** Compagno and co-authors introduce parameterizations for the evolution of debris cover distribution and thickness into a glacier model, applicable on the large regional and global scale. They derive the parameterizations, calibrate parameters based on observations (some of them depending quite strongly on another model), and evaluate the parameterizations using independent observations. They apply the model for projections of glacier evolution in the 21st century in High Mountain Asia.

There is no doubt that the manuscript improves the state-of-the-art of how debris cover is represented (if at all) in glacier mass balance models applicable to the regional and the global scale. The advances presented in the manuscript clearly contribute significantly to our understanding of how relevant debris cover is in shaping the future of glaciers. The manuscript is generally well written and the results are generally presented well, but I also have a relatively large number of minor or technical comments and suggestions.

(ii) There are, however, a few issues with the manuscript that are more substantial. I believe that the authors will be able to address them, and I don't believe that the main conclusions of the manuscript will change. But since they could only be addressed in extensive revisions to the text and/or additional analyses (see below for details), I consider them major.

**[AR 1.00] (i)** We thank the reviewer for the very positive feedback and for the review. **(ii)** Below, we have addressed all issues raised by the reviewer. The manuscript was updated accordingly.

**Major comments**

**[RC 1.01]**- The estimation of debris cover thickness (L91-99) is very unclear to me: (i) why is "observation" in "observations-based mass balances" in quotes? (ii) how is the energy balance model that you apply on each glacier calibrated, ie., and how are the glacier-specific Østrem evaluated? (iii) What is the reasoning behind Eq. 1, and the meaning of the "free parameters" i\_debris and k\_debris? I appreciate that you cannot repeat the manuscript of McCarthy et al. here, but the description needs to be understandable in principle without going to the reference (even if it was accessible to readers, which it is presently not). (iv) Probably I just don't get it, but I am also left a bit puzzled why a temperature-index melt model is applied for the projections if the authors have an energy balance model that can deal with debris cover and is applicable for each glacier, and which the authors trust so much as to not only estimate debris thickness, but additionally glacier-specific Østrem curves.

(v) I have a similar difficulty following L154-166: may the problem is that it remains unclear whether the goal of the equations is to mimic a physical understanding of the debris effect (such that it is possible to explain the "meaning" of the different parameters), or to parameterize the shape of the Østrem curve. Could, e.g., g be called a "melt modification" parameter (or "enhancement factor", as in Fig. S1) for the temperature index parameter? And is the goal of Eq. 3 to create the shape of the Østrem curve for g? (vi) It might be helpful to include an example of a parameterized Østrem curve with labels for the different threshold and critical values of h (i.e., a schematic version of Fig. S1 including the names of the parameters – and potentially 2 or 3 different curves for different parameter values).

(vii) Finally, more needs to be said on the uncertainties of the estimated debris thicknesses. Otherwise, it is very hard to make sense of the relevance of e.g., the thickness differences presented in Fig. 5.

**[AR1.01] (i)** 'Observations' was in quotation marks because the SMB data of Miles et al. are not observations in the traditional sense, i.e. they are not derived from measurements made in the field or directly from satellite imagery. Instead, they are calculated from geodetic mass balance

estimated from satellite data (Brun et al., 2017) by solving the continuity equation for each glacier. In addition to the geodetic mass balance data, this also requires ice thickness (taken from Farinotti et al., 2019) and surface velocities (taken from Gardner et al. 2019). These new estimates of altitudinal resolved glacier mass balance (which we referred to as observations) are then used in McCarthy et al. to calculate debris thickness. Stated differently: the approach by McCarthy et al. uses the estimates as observations that the inversion aims at matching. Since we recognize that the text was somewhat unclear, we reformulated it. In particular, we now avoid the wording "observation" (see iii).

(ii) The energy-balance model in the study by McCarthy et al. (in review) is not calibrated: it is a physical model with physical parameters (e.g. conductivity) for which we use values reported in the literature, and a model that requires meteorological variables as input. We recognize that both the physical parameters and the input meteorological variables have uncertainties. This uncertainty is estimated in a Monte Carlo framework in which both are perturbed within their expected uncertainty ranges. This allows for generating the mass balance data which are in turn used to determine the Østrem curves. The Østrem curves themselves are not validated since the limited observational evidence precludes it. Instead, we validate the calculated debris thicknesses using all available in-situ debris thickness data with satisfactory results (see McCarthy et al., in review). Note that the work by McCarthy et al. is now available on a preprint server ( <a href="https://doi.org/10.31223/X5WW5B">https://doi.org/10.31223/X5WW5B</a>) thus making the full description of the methods directly accessible. We added the missing information in the manuscript at L. 97-117 (see iii for suggested text).

(iii) Equation 1 describes the conceptual relationship between debris thickness and sub-debris melt rates that is represented by the Østrem curve. The free parameters idebris and kdebris are determined for each glacier by fitting a curve of the form given by Equation 1 to (a) the surface mass balance generated using the energy-balance model, and (b) the debris thicknesses used as input to the energy-balance model. We reformulated the entire paragraph, in order to make the procedure clearer.

L. 97-121: 'The debris thickness maps are based on McCarthy et al. (in review), and were obtained through a simplified surface-mass-balance inversion procedure similar to Ragettli et al. (2015) and Rounce et al. (2018). In a nutshell, the inversion procedure iteratively solves for the debris thickness by using an energy balance model. The procedure uses DEMs, glacier ice thickness, surface velocity, debris proprieties, and meteorological forcing data as input, and uses them to calculate ice flux divergence and ice thinning rates. The debris thickness is then adjusted until modelled and observed ice-melt rates agree within a prescribed

tolerance. Due to the physical nature of the procedure, the energy-balance model and the Østrem curves (see below) do not need calibration. The debris thickness maps are however evaluated using a high number of available in-situ data (148007 data points on 13 glaciers) on debris thickness, showing good agreement (see McCarthy et al., in review).

To generate the Østrem curves, in a first step, the energy-balance model was run at randomly chosen points on the surface of each considered glacier, and with debris thicknesses and debris properties randomly chosen within expected physical ranges. These Østrem curves are expressed as:

$$b = \frac{i_{\text{debris}} \cdot k_{\text{debris}}}{h + k_{\text{debris}}} \tag{1}$$

where b is the local surface mass balance, *i*debris and *k*debris are parameters to be determined, and h is the debris thickness (m). Note that equation 1 has similarities with the Hyper-fit model of Anderson and Anderson (2016), and Anderson et al. (2021a, b), although we note that the two approaches differ in the number of parameters and their interpretation. In a second step, the mass balances inferred by Miles et al. (2021) were used together with the fitted Østrem curves (Eq. 1) for each elevation (i.e. assuming that englacial and basal mass balance is negligible) to determine the debris thickness maps used in this study. The so-obtained information represents the supraglacial debris conditions for the period 2000-2016. With the method described above, McCarthy et al. (in review) estimate a mean debris thickness for the debris-covered part of all glaciers in HMA of

0.34m (with an uncertainty between 0.15 and 0.76 m). The uncertainties are asymmetric because surface mass balance is less sensitive to debris thickness as debris thickness increases, and are in line with other studies (e.g. Rounce et al., 2021). For our purposes, the spatially-distributed debriscover information is divided into elevation bands of 10 m whilst the Østrem curves were directly added into our mass balance module (see section 3.1)."

(iv) There are two main reasons for why a temperature-index approach is used for the forward projections: (1) Running a distributed energy-balance model over the whole of HMA and up to the year 2100, would be computationally unfeasible; the energy balance model in McCarthy et al. was run at the point scale, with one point for each elevation band of each glacier, and this alone required a very large computation effort. (2) An energy-balance model requires additional meteorological forcing (e.g. wind, humidity, solar radiation) which may be much more uncertain in future simulations than it is in the meteorological reanalyses.

(v) The goal of Equation 2 is to mimic the relationship between debris thickness and sub-debris melt in a manner that is appropriate for an empirical, temperature index model. We sought for a simple, functional relationship that can be represented in the GloGEMflow framework. As such, the parameters do not have a strict physical meaning.

We hope that with the reformulation of L. 97-121 (cf. 1.01 iii) of our method is clearer. Further, we clarified the meaning of 'g', which is a factor enhancing ablation when debris is present. L. 179:'*The factor g (which acts as a factor enhancing ablation due to debris) depends on* [...]'

(vi) As suggested by the reviewer, we added a figure with different curves for different parameter values, and a visual explanation of Eq. 2 and 3.

Figure 3: 'Schematic of the melt enhancement factor g (dimensionless) as a function of debris thickness for three different glaciers (green lines). The colored, dashed boxes show regions in which the different cases of Eq. 2 and 3 apply.'

(vii) The uncertainty in our debris thickness estimates is moderately high. This is because of the variety of input datasets required by McCarty et al. (in review) and because of our conservative error propagation approach (in which the uncertainty accumulates). For the whole of HMA, we estimate a mean debris thickness of 0.34 m (with an uncertainty of between 0.15 and 0.76 m). We note that similar relative uncertainties are produced by, e.g. Rounce et al, (2021).

We added the overall debris thickness uncertainty (cf. AR 1.01 iii) and the glacier-specific uncertainty in Fig. 1

---

## Author Response (AR2)

**Author's response to the comments received for tc-2021-31**

The following pages contain a point-by-point reply to the comments provided by the editor and the referee that commented our third submission.

Each of the editor's comment (**EC**) and referee's comment (**RC**) is numbered. If a comment contained several points, we numbered them, and address them individually in our author replies (**AR**).

**Editor comments**

**[EC 0.01]** Dear Loris Compagno and co-authors,
**(i)** Your manuscript received two very detailed and critical reviews that both highlighted the novelty and innovation in the treatment of debris in regional/global scale glacier models and the related extensive modelling results/investigations and they confirm that the manuscript would be a very valuable contribution to TC. Although both referees were in general very positive, they also raised, besides the minor technical corrections, a few major issues that should be improved before publication. **(ii)** In brief the most important points were
a) clarification and more details in the methods (in particular the debris thickness data and the parametrized oestroem curve)
b) issue of omitting dynamic debris thickening/thinning (velocity) effects (add at least a discussion on this)
c) better evaluate the role of debris cover inclusion (omitting, explicit, implicit)

As the revisions were rather substantial the revised version went to review again (to one of the previous reviewers), and I myself as the editor, also read and checked the revised version again. The re-review clearly stated the manuscript substantially improvement and almost all major issue were addressed well (see detailed reviewer comments further below), which I generally confirm. However, besides a few very minor points, the referee states that a few more substantial points seemed to remain which concern the used debris thickness datasets of McCarthy and are re-iterated below. I generally confirm the referees view on this and in addition.

**[AR 0.01] (i)** We thank the editor for the very positive feedback and for the comments. **(ii)** Below, we have addressed all issues raised by the editor. The manuscript was updated accordingly.

**[EC 0.02] (i)** I have one further more substantial point**:** i) (raised by rereview, see also list below): The referee is still concerned with the impression in the text that the McCarthy dataset (debris thickness) is based on first principle and does not require any calibration, which I agree with the referee is not true, as there the derivation uses besides the principle of mass conservation for example an EB-model which surely had to make some parameter choices (and probably some calibration). These parameters may be based on literature or have been calibrated (not by you but in the McCarthy paper), thus, the statement that no calibration is needed should be revised and clarified.
**(ii)** I have one more further more structural comment on this paragraph on the explanation of the derivation of the McCarthy debris thickness: you start with saying '…obtained through a simplified surface mass-balance inversion procedure…' which I understand is based on the principle of the mass conservation using remote-sensing data as input (dh/dt, velocities). But you then first mention the EB-model and only afterwards the mass conversation which I find a bit confusing. I would suggest to move the mass conservation principle (using …data) right to the beginning and then later explain the use of the EB-model for calculating debris thickness from the inverted SMB, as this is the order you do it (or am I completely wrong here?). something along the lines of: '…obtained through a simplified surface mass-balance inversion procedure similar to (Ragettli …) that is based on the principle of mass conservation using surface velocities and thinning rates as

input. It then uses an energy-balance model, meteorological data to iteratively solve for debris thickness ....'

**[AR 0.02]**: (i) We agree with the editor and with the reviewer (cf. AR 1.02 a) that this sentence was odd and that it could be misunderstood. Therefore, we reformulated it as follows:
 *L.104-105: 'Due to the physically-based nature of the procedure, the energy-balance model and the Østrem curves (see below) are not explicitly calibrated, but use model parameters that are based on literature value.'*

(ii) We changed the sentence based on the editor's as suggestion:
ll.97-101: *'The debris thickness maps are based on McCarty et al. (2021, preprint), who used a simplified surface mass-balance inversion procedure similar to Ragetti et al. (2015) and Rounce et al., (2018). In a nutshell, the procedure uses the principle of mass conservation to infer on local glacier mass balance from surface velocities and thinning rates, and then iteratively adjusts the debris thickness to ensure consistency between the so-inferred mass balance and the output of an energy-balance model driven by meteorological data. More specifically, the procedure uses DEMs, glacier thickness, [...]'*

**[EC 0.03]** ii) (raised by rereview, see also list below): the referencing to unpublished work of McCarthy et al. (see line 97,...) is an issue and as it is essential for being able to trace the explanation for the debris thickness dataset. As the referee suggests either you wait until the McCarthy paper is accepted (which would likely delay things) or alternatively (which probably makes more sense) you refer to the preprint version link and make clear in the reference that it is preprint.

**[AR 0.03]**: Due to the fact that McCarthy et al. is still under revision, we decided to use the preprint as citation. Both in the references list and in the text, we make clear that we are referencing a preprint.

In references: *'McCarthy, M., Miles, E., Kneib, M., Buri, P., Fugger, S., and Pellicciotti, F.: Supraglacial debris thickness and supply rate in High Mountain Asia, Communications Earth and Environment, https://doi.org/10.31223/X5WW5B, 2021, preprint.'*

In text: *'McCarthy et al. (2021, preprint)'*

**[EC 0.04]** iii) an other point by the editor: the issue raised by referee 2 of neglecting explicit consideration of dynamic debris thickening/thinning. I understand that including such explicit dynamic effects are out of scope of this manuscript and to some degree these effects are indirectly included in the parameterization. This is now well discussed in the discussion chapter (7.2 lines 501-509) which is useful. However, I think it would make the paper much clearer when you would say more explicitly already in the method explanation of the debris evolution parametrization (section 3.2.3) that dynamic thickening or thinning due to velocity changes (spatial or temporal) are not explicitly modelled but implicitly included in the debris parametrization through.....

**[AR 0.04]**: We added this information also in the Methods section:
l.260-262: *'Combined with $b_{z,t}$, the long-term glacier-wide mass balance $B_{(t-9,t)}$ implicitly accounts for ice-dynamical processes, e.g. thickening or thinning due to spatial and temporal changes in ice flow velocity (see section 7.2 for a detailed discussion)'.*

**[EC 0.05]** Overall this manuscript is now close to publication but the few more substantial points above (i to iii) and the mostly minor editing issues listed below (from re-review and the editor) should be addressed by the authors before acceptance of the paper.

I thank the authors for their collaboration and detailed response of how to address the points raised by the referees and congratulate them for the paper.

Andreas Vieli, editor
19 march 2022

[AR 0.05] We thank also the editor for his collaboration and valuable suggestions.

**Minor comments by editor**

[EC 0.06] Line 4: '….and implement IT (not IS) as a module…'

[AR 0.06] Corrected

[EC 0.07] Line 95: delete the comma between 'glacier-specific' and 'Oestroem curves'

[AR 0.07] Corrected

[EC 0.08] Line 236: do you mean 'number of elevation bands h'? rather than amount, because in eqn (6) you use the number symbol.

[AR 0.08] Corrected. Yes, we meant 'number of elevation bands h'

[EC 0.09] Line 254: odd sentence 'It constrained based on observations (see section 4.2).' maybe now redundant?

[AR 0.09] We deleted the sentence as indeed, it was redundant now.

[EC 0.10] Fig. 6 caption: I appreciate that the caption is now shorter and easier to read, but you should still state what the numbers are in the y axis (RGI glacier numbers???) and clarify that the results are in relation to the 'thickness tuning factor' (I assume this is the c_thickness parameter?).

[AR 0.10] We added the missing information in the figure caption. We also changed 'thickness tuning factor' into '$c_{thickness}$', so that it is now consistent with the text (cf. AR 1.04)
Fig. 6: *'Difference between observed and modelled debris thickness for the period ~1981-2008 (circles) using different thickness tuning factor ($c_{thickening}$). Glaciers ID's refer to RGI6.0 (RGI Consortium, 2017)'*

[EC 0.11] Acknowledgments: it would be good to acknowledge the referees input.

[AR 0.11] We added the editor and the referees in the acknowledge:
Ll: 597-598: *'Finally, we are grateful to the Editor Andreas Vieli, and the two reviewers Ben Marzeion and Leif S. Anderson for their numerous constructive remarks and suggestions.'*

**Re-review comments**

[RC 1.01] (i) The authors have done very thorough and comprehensive revisions, and their replies (and changes in the manuscript) have clarified almost all my points. The additions to the manuscript (e.g., new Fig. 3) are also very helpful.
(ii) I am still concerned with the reliance of the manuscript on a yet unpublished paper (see below for my only remaining "major" comment). I don't have a simple solution for this, but I would recommend against publishing a manuscript as long as a key reference is not publicly available.

**[AR 1.01] (i)** We thank the reviewer for the very positive feedback and for the second review. **(ii)** Below, we have addressed all issues raised by the reviewer. The manuscript was updated accordingly.

**Major comment**

**[RC 1.02]** The description of how the debris-cover thickness is generated is much improved, and I thank the authors for making available the McCarthy et al. manuscript.

**(i)** I disagree with the claim that "Due to the physical nature of the procedure, the energy-balance model and the Østrem curves (see below) do not need calibration" (L115 of tracked-changes version). Agreed, the model used in McCarthy et al. is a much more complex model, and it resolves many more physical processes. However, it clearly includes (a relatively large number, I would argue) parameters; be it emissivity of debris, bulk thermal conductivity, debris heat capacity, then the turbulent fluxes which themselves depend on parameterizations, etc. The authors' claim that these parameters are not calibrated but taken from the literature misses the point: if, e.g., the surface roughness length of debris is taken from the literature, then presumably that literature used observations to calibrate the parameterization of a logarithmic wind profile.

Two thoughts on this issue:
**(ii)** (a) My point might be seen as nitpicking. To some extend I would agree, but the risk of readers misunderstanding the authors' claims as implying that the model in McCarthy et al. is based on first principles is considerable, and this could induce a wrong understanding of how models work. Models are always approximations and are always depending on simplifications (or parameterizations - no matter what you call it). There are, however, many different levels of complexity in models, and different complexities are needed for different tasks.

**(iii)** (b) It is a problem that I feel the need to start reviewing another (so far unpublished) manuscript to understand and review this one. Neither do I want this, nor can it be in the interest of the authors. I also see a risk that one paper gets published, and the other one doesn't (or potentially, after substantial revisions).

**(iv)** I therefore still see the dependency on McCarthy et al. as critical, and I don't think the manuscript on hand should be published before the McCarthy et al. manuscript is generally accessible. The authors will need to decide whether to cite the EarthArxiv version (implying they have to rely on a somewhat volatile source) or wait for the publication.

**[AR 1.02]**
**(i)** The sentence was poorly formulated, and we agree with the reviewer and the editor (cf. AR 0.02 a) that the sentence created more questions than answers. We reformulated the sentence into:
L.103-105: *: 'Due to the physically-based nature of the procedure, the energy-balance model and the Østrem curves (see below) are not explicitly calibrated, but use model parameters that are based on literature value.'*

**(ii)** We agree with the reviewer. We hope that with the reformulated sentence (cf. AR 1.02 a), the risk of misunderstanding no longer exists.

**(iii)** We understand the reviewer's concern and fully agree that the situation is unfortunate. When preparing our article, we were not expecting the submission- and review-process of the work by McCarthy to be so slow. Based on the reviews that the article received in the meanwhile, we are however more than confident that the situation in which one paper gets published and the other doesn't will not arise. Moreover, what we know by now is that the methodology by McCarthy won't need substantial revision. Since the publication date of the McCarthy article is not yet set, however, we decided to cite the EarthArxiv preprint version (cf. AR 0.03).

**(iv)** See above and AR 0.03: we have opted to cite the EarthArxiv preprint, clearly labelling it as such.

**Minor comments**

**[RC 1.03]** Description of i_debris and k_debris: please add units to the parameter values so that eq. 1 works out.

**[AR 1.03]**

The units of $i_{debris}$ and $k_{debris}$ are 'm w.e./a' and 'm', respectively. The manuscript was updated according:

ll. 113-114: *'where b is the local surface mass balance (m w.e. a⁻¹) , h the debris thickness (m) and $i_{debris}$ (m w.e. a⁻¹) and $k_{debris}$ (m) are glacier-specific calibration parameters without specific physical meaning.'*

**[RC 1.04]** Fig. 6: Since uncertainties of the debris cover thickness are available (Fig.1), it would be good if significant deviations between model and observations would be emphasized (e.g., by adding a black ring). This will help understanding the range of suitable values of c_thickening (also, please make the x-axis label consistent with the variable name in the text).

**[AR 1.04]** We changed the x-axis label into '$c_{thickening}$' to ensure consistency with the text.

[Figure]

**[RC 1.05]** Fig. R1: majority of glaciers in WGMS clean ice: good point. But apparently, there are debris-covered glaciers in the sample, otherwise the numbers should be exactly the same. Do a meaningful number of data points remain if the clean-ice glaciers are filtered out? I really think the evaluation of how the new parameterization affects model performance would be very helpful, and could be a strong point of the paper!

**[AR 1.05]** We went manually trough all HMA glaciers present in the WGMS database, and there are only two candidate which are partially debris covered: RGI6.0-14.15990 (Fig. R1a) and RGI6.0-13.43277 (Fig. R1b). The remaining glaciers are not debris covered. However, both glaciers (1) do not have a fully debris covered tongue and (2) have WGMS mass balance measurements only for elevation bands (no point measurements available). Since we don't know if the WGMS elevation-band data refers to the clean-ice or debris-covered part of the tongue, we feel that such a comparison between observed and modelled mass balance could be misleading.

[Figure]

**Figure R1:** satellite image of (a) RGI6.0-14.15990, (b) RGI6.0-13.43277. Source: (a) Google Earth, (b) Sentinel Hub.

**[RC 1.06]** -Fig. S3: remove "annual" from title.

**[AR 1.06]** Corrected

**[RC 1.07]** -AR1.21(i) the colored numbers still seem to be in the plot?

**[AR 1.07]** This was our mistake. We now deleted the colored numbers.

**[RC 1.08]** -AR1.23: Sorry to insist here, but it has also not been shown that the explicit treatment of debris-cover leads to a more "adequate" capture of quantities like local mass balance, glacier length, or runoff. To make this point, e.g., for the mass balance, an improvement of the model performance on debris covered glaciers (cf. Figs. S2/R1) would have to be shown.

**[AR 1.08]** We changed the sentence into:
L480-482: *'Indeed, quantities such as the local mass balance, the glaciers' ice flow velocity and mass turnover, the glacier's length change or water runoff, are different when explicitly accounting for supraglacial debris and its temporal evolution.'*